# Classification of Alpine South Foehn based on five years of km-scale analysis data

Jansing Lukas[1], Papritz Lukas[1], Dürr Bruno[2], Gerstgrasser Daniel[3], and Sprenger Michael[1]

[1]Institute for Atmospheric and Climate Science, ETH Zürich, Zurich, Switzerland
[2]Sunergy GmbH, Degersheim, Switzerland
[3]MeteoSwiss, Zurich, Switzerland

**Correspondence:** Lukas Jansing (lukas.jansing@env.ethz.ch)

**Abstract.** It has long been recognized that a rich variety of Alpine South Foehn flavors exists that is related to varying flow conditions above crest level, the presence and intensity of orographic precipitation on the Alpine south side, and the Po Valley stratification. This study presents a systematic five-year climatology of different Foehn types. The classification relies on 2'329 Foehn hours, which are diagnosed using a station-based Foehn index for Altdorf in the Swiss Reuss Valley. Operational analyses at 1-km horizontal resolution are employed to classify Foehn hours with a decision tree that is based on Foehn forecasting experience. Mean wind direction and speed around Altdorf are considered to differentiate between three main Foehn types (*Deep Foehn*, *Shallow Foehn*, *Gegenstrom Foehn*). In addition, upstream precipitation and its extent beyond the Alpine crest are used to distinguish three *Deep Foehn* subtypes (*Dry Foehn*, *Moist Foehn*, *Dimmer Foehn*).

The main Foehn types differ distinctively in the synoptic conditions over the Alps. During *Deep Foehn*, pronounced southwesterlies ahead of an upper-level trough induce upstream orographic precipitation. *Shallow Foehn*, in turn, is associated with cross-Alpine temperature differences that provoke a gap flow. The *Gegenstrom Foehn* type is also restricted to major gaps, but a strong westerly flow prevails above crest level. The *Deep Foehn* subtypes primarily differ in terms of the upper-level trough. While a weaker trough and the influence of an upper-level ridge over the Mediterranean inhibit precipitation (*Dry Foehn*), a deeper trough which is closer to the Alps induces stronger crest-level winds and intense precipitation on the Alpine south side (*Dimmer Foehn*). The different Foehn types are found to strongly affect the local characteristics at Altdorf, which are investigated using station measurements. Furthermore, the occurrence of a particular Foehn type at Altdorf impacts the probability of concurrent Foehn winds at other locations on the Alpine north side.

Backward trajectories from Altdorf are calculated for each of the Foehn hours and used to define three clusters of air parcels depending upon their upstream thermodynamic evolution. Trajectories in cluster 1 are diabatically heated and transported within a low-level easterly barrier jet in the Po Valley prior to their ascent to crest level. They constitute the main precipitating airstream and, hence, are of key importance for *Moist Foehn* and *Dimmer Foehn*. Cluster 2 and 3 trajectories are subject to weak diabatic heating or even diabatic cooling. They originate from southerly to southwesterly regions and from either slightly below or above crest level. Accordingly, these air parcels are associated with little to no precipitation and as such, they take a key role for *Dry Foehn*, *Shallow Foehn* and *Gegenstrom Foehn*. Furthermore, these three Foehn types feature a pronounced stable layer over the Po Valley, which, to some extent, inhibits air parcels to ascend from lower levels.

In summary, the study introduces a systematic classification of South Foehn using state-of-the-art data sets. It concludes by setting the new classification into a historic context and revisiting the rich body of literature with respect to different Alpine South Foehn types. In particular, analogies and discrepancies to the existing conceptual models of 'Swiss Foehn' and Austrian Foehn' are discussed.

## 1 Introduction

Foehn, originating from the latin *favonius* (e.g., Sprenger et al., 2016), represents a generic term for downslope winds and windstorms in the lee of mountains. These strong and often gusty winds are associated with a distinct warming and relative drying of the air on the lee-side of the orographic barrier (e.g., Richner and Hächler, 2013). The Alpine Foehn profoundly impacts societies in the affected regions. For example, Foehn occurrence enhances the risk for forest fires (Zumbrunnen et al., 2009; Wastl et al., 2013). At the same time, the Foehn-induced microclimate can benefit agricultural production, e.g., viticulture in Foehn valleys (Walker and Ruffner, 1998). Accordingly, this archteypal wind phenomenon has already piqued the curiosity of the 19th century Swiss naturalists. For example, Escher von der Linth attributed the occurrence of ice ages to the lack of Foehn winds during these periods (the Sahara theory; see, e.g., Lehmann, 1937). Additional, sometimes fierce, disputes among scientists concerned the reason for the exceptional warmth of the Foehn air, as well as the driving mechanism for the descent into the Foehn valleys (see, e.g., Steinacker, 2006, for a summary of different Foehn theories).

Alpine South Foehn events are associated with characteristic synoptic situations. The generic situation is that an upstream upper-level trough induces southwesterly to southerly large-scale flow towards the Alps. Meanwhile, at the surface, an extra-tropical cyclone and its attendant frontal system approach Central Europe. As a result, a pressure gradient builds up across the Alps and Foehn winds break through on the Alpine north side. Detailed descriptions of the synoptic evolution of so-called *Deep Foehn* events can be found in the literature (e.g., Burri et al., 1999; Richner et al., 2006; Hächler et al., 2011; Jansing and Sprenger, 2022). With respect to *Deep Foehn*, Burri et al. (1999) differentiates between 'pre-frontal Foehn', where short-lived Foehn episodes are terminated by rapidly propagating cold fronts (e.g., Hächler et al., 2011), and long-lasting Foehn situations related to stationary upper-level troughs ('trough Foehn' situations), where the upstream trough over Western Europe can remain quasi-stationary for up to several days (e.g., Jansing and Sprenger, 2022).

However, even early on, scientists also recognized the existence of different Foehn varieties and different stages within a Foehn event. In fact, Hann (1885) already emphasized that Foehn can occur without upstream precipitation (in our terminology, we refer to it as *Dry Foehn*), and oftentimes Foehn breakthrough precedes the formation of orographic precipitation. Ficker (1910) divides the typical Foehn event into two stages: During the 'anticyclonic stage', while Foehn winds already established on the Alpine north side, air does not ascend on the south side. Instead, the free troposphere is separated from the air below by an inversion layer, above which southerly winds and very low humidity values are observed. While a Foehn event can terminate at this stage, it is usually followed by the 'stationary stage', where upstream precipitation starts to form and the characteristic Foehn wall establishes. According to Kuhn (1984), Billwiller first introduced the term 'anticyclonic Foehn', but interpreted it as the subsidence of air within an anticyclone situated over the Alpine ridge. For this reason, Gubser (2006) does

not classify this Foehn type or stage as a Foehn wind. On the other hand, in most publications, 'anticyclonic Foehn' or the 'anticyclonic stage' is interpreted as the Foehn phase without upwind precipitation and little to no cloud formation (e.g., Burri et al., 1999; Gerstgrasser, 2017), often followed by the 'stationary stage' or 'cyclonic stage' (see also in Frey, 2007). Another example following this evolution is the 'once-in-a-century Foehn event' ('Jahrhundertföhn') from 7th and 8th November 1982 as described by Frey (1984), where record-breaking cross-Alpine pressure differences of 24 hPa were observed.

Another Foehn variety, primarily discussed by Swiss Foehn researchers, concerns the so-called 'dimmerfoehn', in the following referred to as *Dimmer Foehn* (Richner and Dürr, 2015). Streiff-Becker (1933) describes the local meteorological conditions associated with *Dimmer Foehn*: The Foehn wall extends unusually far beyond the main Alpine crest. As a consequence, the otherwise dry Foehn regions appear hazy due to the presence of humid and cloudy air. Oftentimes, the humid air precipitates into the Foehn valleys and induces evaporative cooling, which strongly reduces the dewpoint depression within the valley atmosphere (Richner and Hächler, 2013). Yet another peculiarity with respect to *Dimmer Foehn* is related to the shift in its northward extent: As the upper part of the Foehn valleys is situated within the humid air being advected over the main crest, the Foehn reaches the ground further down-valley compared to the normal *Deep Foehn* (Streiff-Becker, 1947). According to Streiff-Becker (1947), the Foehn also advances unusually far north. While *Dimmer Foehn* is regarded as a rare Foehn variety with a local lee-side impact, the synoptic conditions nonetheless deviate from *Deep Foehn*. The upstrem low pressure system tends to be closer to the Alps and, as a result, large-scale southwesterlies or southerlies above crest level are stronger. Streiff-Becker (1947) also emphasizes the frequent presence of secondary lee-cyclones during such events. To this point, however, no systematic evaluation of the characteristic synoptic to mesoscale conditions during *Dimmer Foehn* has been performed.

The aforementioned Foehn types share a common characteristic, namely the presence of an upstream disturbance inducing flow with a southerly component towards the Alps. Besides, however, it has also been noticed, primarily by Austrian Foehn researchers, that South Foehn can likewise establish without the presence of a pronounced cross-Alpine flow component above crest level. In his textbook, Hann (1901) briefly mentions the occurrence of Foehn due to smaller cross-Alpine pressure differences, when no southwesterly winds are present in the Alpine foreland. Ficker (1931) describes the development of Foehn winds, while above crest-level winds are weak and, sometimes, even from a northerly direction. To our knowledge, the first appearance of the term *Shallow Foehn* occurs in Kanitscheider (1932), as he depicts several Foehn cases with a limited vertical extent of the southerly winds in the Innsbruck region, whereas weaker westerlies prevail above a certain level. Seibert (1990) defines *Shallow Foehn* as "... a foehn with all typical properties in the valleys, but without pronounced southerly wind component at upper levels (above crest height).". Applying a range of criteria, roughly 10% of all Foehn hours in Innsbruck are classified as *Shallow Foehn*. However, it is unclear how frequently this Foehn type occurs in the Swiss Alps. Additionally, a mechanistic explanation is included in Seibert (1990): *Shallow Foehn* is regarded as a result of cross-Alpine temperature and pressure differences of the contrasting low-level air masses on the Alpine north and south side, respectively. These, in turn, induce a hydraulic 'compensation flow' through major gaps along the Alpine transect. In fact, Mayr and Armi (2008) extend this line of reasoning even further, by arguing that the presence of north-south temperature differences are a necessary condition for the formation of *Shallow Foehn* as well as *Deep Foehn*, while a southerly flow component merely imposes a modification to the conditions responsible for Foehn formation. They link the build-up of cross-Alpine temperature gradients to the advection of

cold and warm air around the eastern and western edge of the Alps, respectively. The initial driver of the temperature advection is associated with the passage of a synoptic ridge. Seibert (1990) emphasizes that the warm air advection on the Alpine north side, building up the low-level cross-Alpine temperature differences, happens prior to the approach of the upstream trough. Therefore, *Shallow Foehn* is regarded the natural predecessor of *Deep Foehn*.

Güller (1977) describes a Foehn event, where the synoptic conditions neither led to southerlies nor weak winds above crest level, but a strong, northwesterly jet prevailed above the Alps. Such synoptic conditions are typically associated with eastward moving storms on the Swiss Plateau (see Fig. S2a for orientation), but not with the formation of Foehn in the respective valleys. During this event, the cross-Alpine pressure differences mainly built up due to a strong pressure drop on the Alpine north side, related to the rapid approach of a secondary cyclone towards Central Europe. Accordingly, Burri et al. (1999) classifies this Foehn variety as a special type of pre-frontal Foehn, occurring ahead of warm fronts. On the other hand, as the southerly winds are restricted to below crest levels, it can also be classified as an extreme case of *Shallow Foehn* (Seibert, 1990; Sprenger and Schär, 2001). Gerstgrasser (2017) also mentions that this Foehn type (named 'Gueller Foehn' or, in this study, *Gegenstrom Foehn*) can be considered a special variety of *Shallow Foehn*. Yet, from a mechanistic point of view, the cross-Alpine pressure differences build up differently, and the associated synoptic conditions are strikingly different than for *Shallow Foehn*. Overall, there is barely any literature available with respect to this Foehn type, and, to the authors knowledge, no climatological assessment has been done so far.

In the late 90s and early 2000s, research on the Alpine South Foehn culminated in the Mesoscale Alpine Programme (MAP; Bougeault et al., 2001). During MAP, two different foci with respect to Foehn were targeted: One focus of MAP was on the local-scale aspects of Foehn dynamics in the widely branched Rhine Valley (Drobinski et al., 2007). In alignment with the earlier studies, a pronounced case-to-case variability was noticed, e.g. with respect to the presence and intensity of upstream orographic precipitation. For a detailed description of the synoptic conditions associated with Foehn events in the Rhine Valley during MAP, it is referred to Richner et al. (2006). The other Foehn focus of MAP concerned research along the Austrian Brenner transect (Mayr et al., 2007). There, the primary focus was to investigate *Shallow Foehn* as a type of gap flow, whereby the phenomenon was studied in the context of hydraulic theory. MAP publications usually define *Shallow Foehn* as a Foehn with either westerlies or weak winds above crest level (e.g., Gohm and Mayr, 2004).

Recent publications continue to emphasize the contrasts between a 'Swiss' and an 'Austrian Foehn' type. This distinction is, at least to some extent, motivated by the fact that Austrian Foehn events are only accompanied by upstream precipitation in about 50% of the time, whereas Swiss Foehn events are more often associated with precipitation (Seibert, 1990; Steinacker, 2006; Würsch and Sprenger, 2015). By combining high-resolution modelling with online trajectories, Miltenberger et al. (2016) compare the air mass origin and the thermodynamics of air parcels for a *Dry Foehn* and a *Moist Foehn* event in the Rhine Valley. Besides evident differences in the latent heating contributions to Foehn air warming, *Dry Foehn* air parcels also stem from considerably higher altitudes over the Po Valley compared to *Moist Foehn* air parcels, which, partly, ascend from near-surface levels to the Alpine crest. This contrast, as highlighted by the two case studies, also emerges from the climatological study conducted by Würsch and Sprenger (2015): Using a reanalysis data set at 7 km horizontal grid spacing, they present the first climatology of backward trajectories for Swiss Foehn (Altdorf) and Austrian Foehn (Ellboegen). Altdorf trajectories tend

to ascend more steeply, and are more frequently associated with precipitation. However, the study stresses that Austrian Foehn can likewise occur in Switzerland and vice versa. While the Foehn hours are grouped depending on the minimum altitude of the trajectories over the Po Valley, they refrain from an explicit characterisation of different Foehn types for Altdorf. All of the recent Alpine South Foehn studies (Würsch and Sprenger, 2015; Miltenberger et al., 2016; Jansing and Sprenger, 2022) invoke the Lagrangian perspective to highlight the complex upstream flow pattern. The low-level easterly flow over the Po Valley is usually decoupled from the southwesterlies aloft (see also Rotunno and Houze, 2007), which leads to the formation of distinct airstreams contributing to the Foehn flow. There are, to this point, no studies that link these type of upwind airstreams to different Foehn types in a climatological context.

In essence, two centuries of Foehn research have illustrated that there exist different Alpine South Foehn types. These are oftentimes viewed as different, characteristic stages of a single event, but can just as well occur individually. Differences between 'Swiss Foehn' and 'Austrian Foehn' have been highlighted from a Eulerian and a Lagrangian perspective. Most of this literature is, however, focusing on individual case studies. Especially, no climatological investigation with respect to different Swiss Foehn types has been conducted so far. Here, we present the first such analysis for Foehn in Altdorf. The specific questions we address are:

1. Are the different Foehn types mentioned in the literature distinguishable in terms of the synoptic conditions?
2. How does the vertical structure of the atmosphere differ between the different Foehn types on the Alpine scale?
3. Where and from which altitude does the Foehn air originate?
4. What is the typical thermodynamic evolution of Foehn air parcels for the different Foehn types?
5. What is the meteorological impact of the different Foehn types at Altdorf and other Swiss Foehn locations?

In order to tackle these research questions, we make use of the operational forecasting system employed at MeteoSwiss, the Swiss Federal Office of Meteorology and Climatology. It allows us to investigate an extensive five-year period using analyses. The data set, including 3D meteorological fields for all Foehn hours in Altdorf, offers unprecedented opportunities to study mesoscale characteristics of South Foehn types in the Alps.

In the remainder of the paper, after having presented the main data sets and methodological approach (Section 2), the first two research questions are addressed by the means of an analysis-based Foehn classification scheme (Section 3) and a composite analysis of the synoptic and Alpine-scale conditions during the different Foehn types (Section 4). The Foehn air origin and thermodynamic evolution with respect to the different Foehn types are dissected using backward trajectories calculated for each of the 2'329 Foehn hours (Section 5). Afterwards, the local-scale impact at Altdorf and other Foehn locations is scrutinized using observations and Foehn index data from a multitude of stations (Section 6). Finally, Section 7 discusses some aspects with respect to the existing literature and Section 8 summarizes the key results of the study.

## 2 Data and methods

### 2.1 Station measurements and Foehn index

On the one hand, station measurements of the MeteoSwiss automatic meteorological network are used to characterise the local meteorological conditions at Altdorf (Section 6). On the other hand, they serve to identify Foehn hours at Altdorf (Section 3) and all other sites listed in the Supplement (Table S1). The operational real-time identification of Foehn at MeteoSwiss sites is based on an objective Foehn index (Dürr, 2008). For the work presented here, data from an enhanced offline version of the Foehn index maintained by the Alpine Research Group Foehn Rhine Valley/Lake Constance (AGF; www.AGFoehn.org) is used. The most important differences between the enhanced and the operational Foehn index are:

- usage of Grand St. Bernard (2472 m AMSL, abbreviated as GSB) as the reference site at the main Alpine crest instead of Gütsch (2286 m AMSL), which is used as backup site in case of data outage
- extension of the Foehn index for *Dimmer Foehn* and *Gegenstrom Foehn*
- inclusion of Foehn probability $F$ for comparison with numerical weather prediction (NWP) model data

*Dimmer Foehn* and *Gegenstrom Foehn* are identified by testing the station measurements for strong winds from the site-specific Foehn wind sector, which are either more humid or potentially colder than the site-specific Foehn thresholds for relative humidity and for the potential temperature difference to the Alpine main crest site, respectively (Dürr, 2008). *Dimmer Foehn* situations with northwesterly winds at Jungfraujoch (3582 m AMSL) are denominated as *Gegenstrom Foehn* in the enhanced Foehn index.

The Foehn probability $F$ [%] of the enhanced Foehn index is calculated as follows:

$$F = 30\left(\frac{(\theta_s - \theta_{\mathrm{GSB}}) - \Delta\theta_t}{\Delta\theta_m - \Delta\theta_t}\right) + 30\left(\frac{ff_s}{ff_m}\right) + 40\left(\frac{UU_t - UU_s}{UU_t - UU_m}\right), \quad (1)$$

where $\theta$ denotes the potential temperature [K], $\Delta\theta$ the potential temperature difference between the local and the Alpine crest site [K], $ff$ the wind velocity [m s$^{-1}$] and $UU$ the relative humidity [%]. $s$ denotes the local Foehn site, $t$ the site-specific Foehn index threshold and $m$ the mode of the unimodal distribution during Foehn periods with raised site-specific thresholds to obtain the $UU$ threshold (Dürr, 2008). $F < 0\%$ is set to 0% and $F > 100\%$ to 100%.

For the study, the Foehn probability $F$, which is given in 10 min intervals, needs to be converted into a binary Foehn timeseries at hourly resolution. When the mean $F$ within a symmetric interval of [-30 min,+30 min] is larger than or equal to 50%, the respective hour is considered a Foehn hour. The aggregation results in 2'329 Foehn hours during the considered time period (29 Oct 2015–29 Oct 2020).

### 2.2 COSMO analysis data

The Consortium for Small-scale Modeling (COSMO) mesoscale numerical model solves a non-hydrostatic formulation of the governing thermo-hydrodynamical equations of the atmosphere (Steppeler et al., 2003). Subgrid-scale processes, including

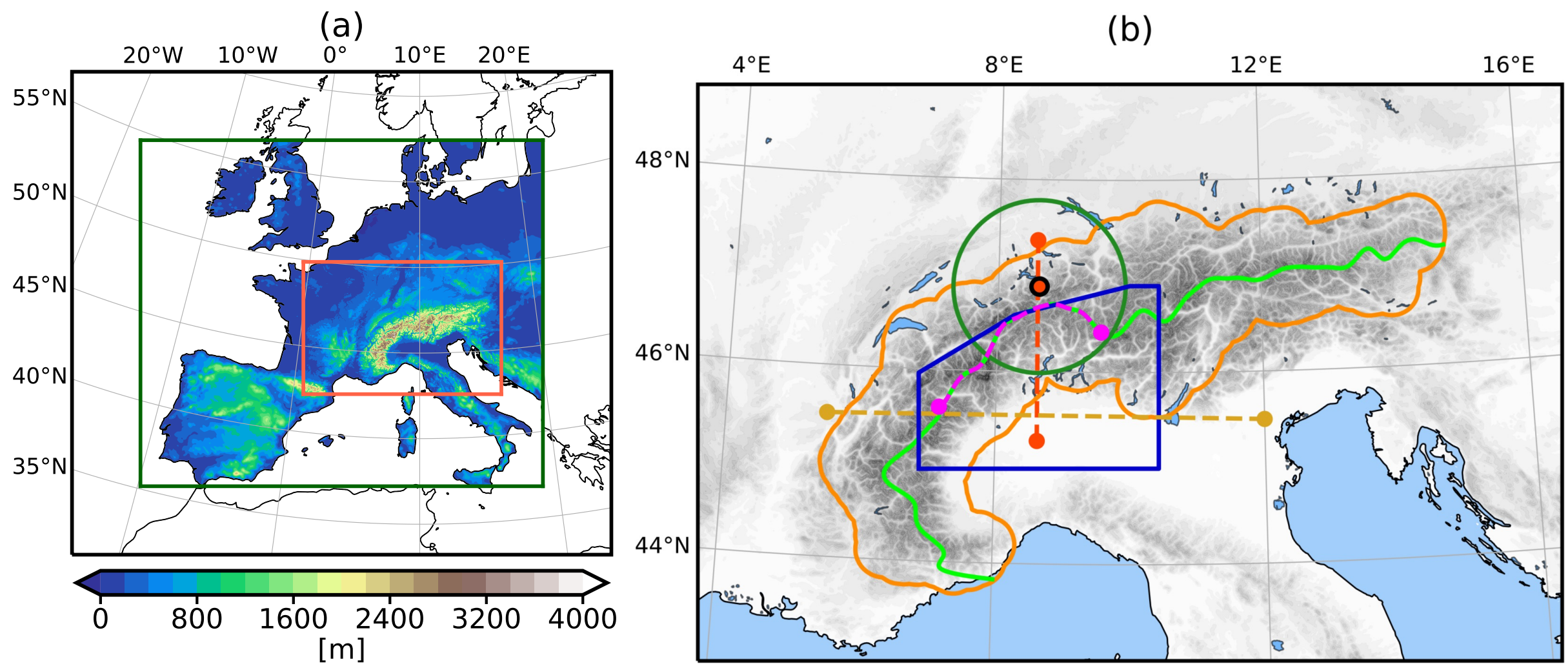

**Figure 1.** (a) Domains and topography of the COSMO-7 (green) and the COSMO-1 (red) analyses, respectively. (b) Polygon around the Alps and the crestline (dark orange and lime). Altdorf is highlighted by a red dot with a black edge. Further indicated are a circle for the wind feature extraction (green) and a polygon for the precipitation feature extraction (dark blue). The cross-Alpine cross section, the along-crest cross section and the upstream Po Valley cross section are marked as red, pink and golden dashed lines, respectively.

cloud microphysics, radiative transfer, turbulence and convection (for coarser resolution) are parametrized. Aside from NWP at the convective scale (Baldauf et al., 2011), the COSMO model also comprises a data assimilation module. At MeteoSwiss, COSMO is used for operational forecasting at several resolutions and for generating atmospheric analyses. This study employs two analysis data sets from MeteoSwiss, namely COSMO-7 (6.6 km horizontal grid spacing) and COSMO-1 (1.1 km horizontal grid spacing). The four-dimensional data assimilation in COSMO, as used for the time period considered, applies an observational nudging technique refined for complex terrain (Schraff, 1997). The assimilated data includes radiosoundings, aircraft and ship data, wind profiler and station measurements (surface pressure, 2-m dew point and wind below 100 m AMSL). Furthermore, radar data is assimilated using a latent heat nudging scheme (Stephan et al., 2008). The assimilation is performed in six-hourly and three-hourly cycles for COSMO-7 and COSMO-1, respectively. While COSMO-7 has been operational since the early 2000s, MeteoSwiss is operating COSMO-1 since March 2016 (pre-operational since autumn 2015). This enables us, for the first time, to systematically analyze and classify Foehn hours during five years on the Alpine scale at 1 km spatial resolution.

For all the Foehn days which are identified (all days with at least one Foehn hour in the considered time period) and the days prior to allow for the calculation of backward trajectories, hourly 3D and surface fields are extracted from the MeteoSwiss archive for COSMO-1 and COSMO-7 analyses.

## 2.3 Trajectory calculations

Air parcel trajectories are frequently used in Foehn research to assess the origin and thermodynamic evolution of the Foehn air in various regions of the world (e.g., Elvidge and Renfrew, 2016; Kusaka et al., 2021), as well as the Alps (e.g., Würsch and Sprenger, 2015; Miltenberger et al., 2016; Saigger and Gohm, 2022). For the Lagrangian analysis based on COSMO-1 (Section 5), we calculate backward trajectories using the Lagrangian Analysis Tool LAGRANTO (Wernli and Davies, 1997; Sprenger and Wernli, 2015). Air parcels are released during each of the 2'329 Foehn hours and at every grid point within a 10 km circle around Altdorf (8.62°E / 46.89°N). The vertical distribution ranges from 20 m AGL up to 3 km with a vertical spacing of 100 m, which results in 4'241 trajectories for each Foehn hour and a total of 9'877'289 trajectories calculated. The internal time step for the integration of the trajectory equation is set to 5 min, while output is saved in 10 min intervals. Standard prognostic fields (temperature, pressure, wind components, specific humidity, five hydrometeor categories), as well as surface precipitation and the topography are traced along the trajectories.

While the analysis is inherently closer to observations compared to a freely running NWP model, a Foehn flow does not necessarily establish in COSMO-1 during each of the 2'329 Foehn hours and in the entire column of the valley atmosphere. To ensure that only trajectories arriving within a Foehn flow are considered for further analysis, air parcels need to fulfill two criteria: First of all, they need to be subject to a southerly wind at their time of release (wind direction between 90°–270°); secondly, they have to intersect the Alpine crestline (Fig. 1b). Together, these criteria reduce the sample size to 7'776'604 trajectories.

Offline trajectories calculated with hourly input wind fields can significantly deviate from online trajectories (Miltenberger et al., 2013) and offline trajectories with higher temporal input frequency (Schär et al., 2020; Saigger, 2021). This is especially true in regions where the flow is substantially distorted and highly transient owing to the complex terrain. Thus, we have to refrain from explicitly assessing the pathways of the air parcels from crest level into the Foehn valleys and from studying the mechanisms that lead to the descent. Instead, we restrict our analysis to the trajectory segments upstream of their crest intersection (Fig. 1b). There, we are confident that the temporal resolution of the input fields from the NWP model suffices to study mean airstream properties.

# 3 Foehn classification scheme

Based on the available literature on different South Foehn types, we aim to distinguish between three main Foehn types related to distinctively different synoptic situations, and hence, crest-level flow conditions. The most typical situation, where the Foehn flow is accompanied by a deep layer of southerly or southwesterly winds (e.g., the first phase of Burri et al., 1999), is referred to as *Deep Foehn*. However, as elaborated in the introduction, Foehn can occur under strikingly different synoptic conditions. We separate between two additional main classes with either weak crest-level winds, i.e. weak synoptic forcing in the Alpine region (e.g. Gerstgrasser, 2017), or an eastward flow regime (e.g. Güller, 1977). These two types are denoted as *Shallow Foehn* and *Gegenstrom Foehn*, respectively. In previous literature, both weak winds and westerlies at crest level are frequently summarized under the term *Shallow Foehn* (e.g. Gohm and Mayr, 2004). However, since the former two flow regimes are

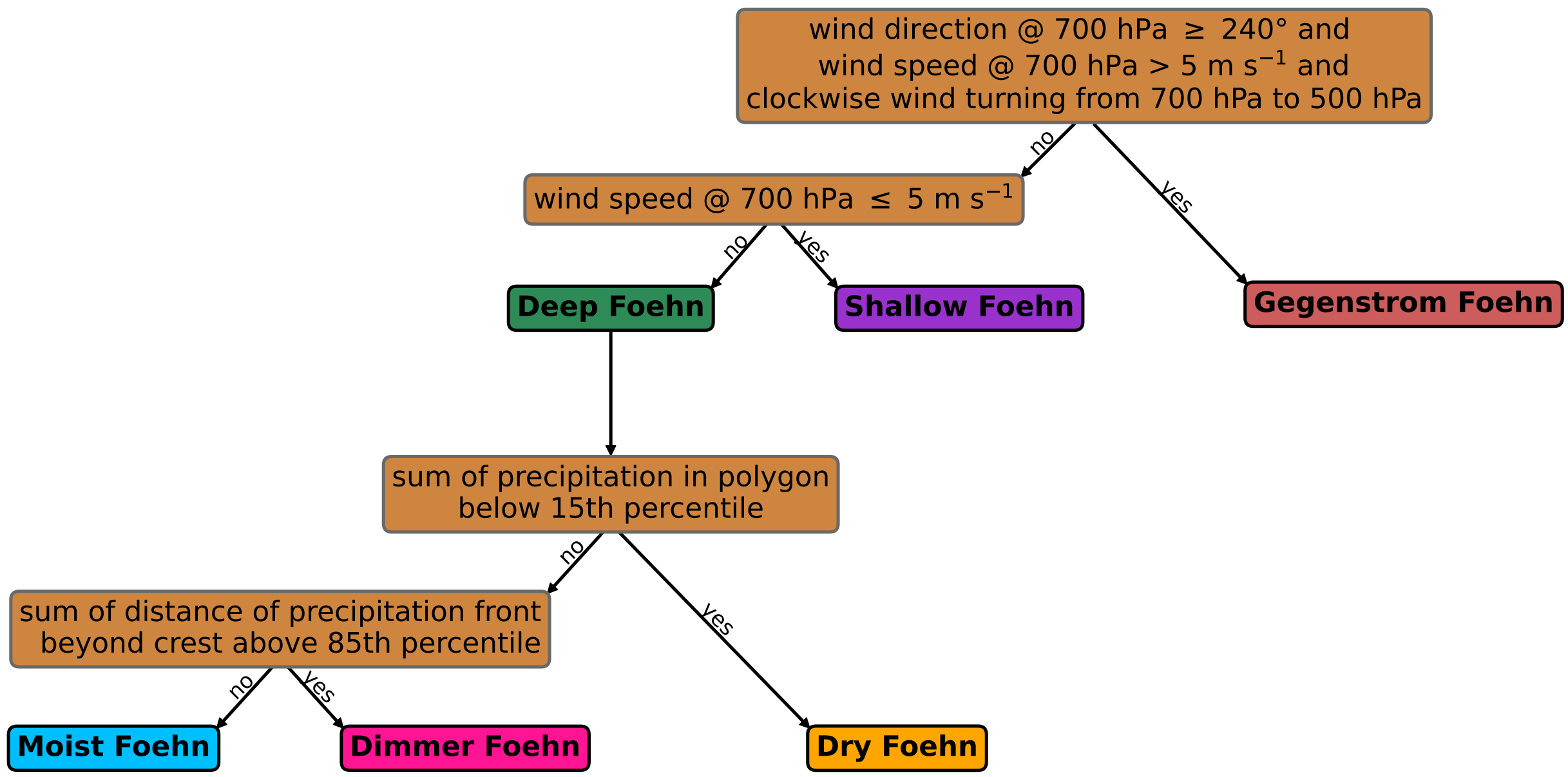

**Figure 2.** Sequential decision tree to classify the Foehn hours into three main Foehn types and three *Deep Foehn* subtypes.

associated with considerably different synoptic situations over Central Europe, as will become evident in the following, we differentiate between them and maintain the term *Shallow Foehn* for cases with weak crest-level winds. Accordingly, we label the zonal regime as *Gegenstrom Foehn*, a term introduced by the AGF (see Section 2.1).

The differentiation of the main Foehn types is done by following a sequential decision tree based on subjective thresholds for the wind direction and speed (Fig. 2). To this end, mean wind speed and direction on 700 hPa and 500 hPa are extracted from the COSMO-1 analysis (see Fig. S2a in the Supplement for a distribution of these quantities for all Foehn hours) in a region centered around Altdorf (green circle in Fig. 1b). Foehn hours with a rather zonal flow at crest level (wind direction between 240° and 360°, i.e. westsouthwest and north) of considerable magnitude (mean wind speed at 700 hPa exceeding 5 m s$^{-1}$) and an anticyclonic wind shear between 700 hPa and 500 hPa are labelled as *Gegenstrom Foehn*. Foehn hours with weak winds at crest level (mean wind speed at 700 hPa below or equal to 5 m s$^{-1}$) are categorized as *Shallow Foehn*, and the remainder is specified as *Deep Foehn*.

*Deep Foehn* can vary substantially with respect to the amount and extent of the accompanying orographic precipitation. In fact, it can occur without any precipitation on the southern side of the Alps (e.g., IOP1 of MAP with 0 mm precipitation in Lugano, as discussed in Drobinski et al., 2007). In stark contrast, other *Deep Foehn* events can be associated with intense upstream precipitation and, in some cases, the transient advection of precipitation beyond the Alpine crest into the northern Foehn regions. The latter is referred to as *Dimmer Foehn* (e.g, IOP2 of MAP according to Richner et al., 2006). Hence, it is reasonable to further subdivide *Deep Foehn* according to the characteristics of the precipitation field (see Fig. S2b for a distribution of the quantities for the *Deep Foehn* subtypes). Extending the decision tree (Fig. 2) further, at first, the spatially summed precipitation within a region (see dark blue polygon in Fig. 1b) extending from the Alps to the central and western Po

**Table 1.** Table providing the Foehn hours for the main Foehn types, their relative occurrence frequency and seasonality.

| | **All Foehn** | **Deep Foehn** | **Shallow Foehn** | **Gegenstrom Foehn** |
|---|---|---|---|---|
| hours | 2329 h | 2080 h | 147 h | 102 h |
| fraction | 100% | 89.3% | 6.3% | 4.4% |
| DJF | 19.8% | 19.2% | 0.0% | 61.8% |
| MAM | 48.4% | 46.8% | 87.1% | 25.5% |
| JJA | 5.1% | 5.0% | 4.8% | 6.9% |
| SON | 26.7% | 29.0% | 8.2% | 5.9% |

Valley (see Fig. S1a for orientation) is extracted from the analysis. It is notable that, while few hours actually exhibit 0 $kg\,h^{-1}$ precipitation, the distribution of the accumulated precipitation is heavily long-tailed and the subjectively chosen threshold is small compared to the median ($8.96{\times}10^9$ $kg\,h^{-1}$). All Foehn hours below the 15th percentile of the precipitation distribution ($3.05{\times}10^8$ $kg\,h^{-1}$), which are also categorized as *Deep Foehn*, belong to the *Dry Foehn* subtype. The remaining *Deep Foehn* hours are categorized with respect to the precipitation extent beyond the Alpine crest line. For an illustrative example the reader is referred to the Supplement (Fig. S3). The procedure is done as follows: Within the crest segment bounded by the light blue circle in Fig. S3, the extent of the precipitation beyond the crest line is identified for each grid longitude by detecting the point north of the crest line where the precipitation falls below $1{\times}10^{-6}$ $mm\,h^{-1}$ for the first time (light red line in Fig. S3). The resulting line is slightly smoothed (dark red line in Fig. S3) and the distance beyond the crest line is calculated for each point of the line. This distance metric is then summed up and all the cases where it exceeds the 85th percentile, and which are not categorized as yet, are labelled as *Dimmer Foehn*. The remaining Foehn hours are labelled *Moist Foehn*, corresponding to the archetypal *Deep Foehn* subtype.

In summary, this classification scheme results in three main Foehn types (*Deep Foehn*, *Shallow Foehn*, *Gegenstrom Foehn*) and three *Deep Foehn* subtypes (*Dry Foehn*, *Moist Foehn*, *Dimmer Foehn*). Their relative occurrence frequency compared to all Foehn hours and their seasonality are displayed in Tables 1 and 2. The overall Foehn hours and their seasonality approximately correspond to the climatological values as identified by Gutermann et al. (2012), whereas the overall Foehn frequency is slightly reduced (466 $h\,year^{-1}$ compared to 483 $h\,year^{-1}$ for 1955–2008 in Gutermann et al., 2012). The correspondence indicates that the five-year Foehn timeseries used in this study constitutes a time period that represents the climatological Foehn occurrence and seasonality in Altdorf reasonably well. The frequency of Alpine South Foehn peaks in spring and, to a smaller extent, in autumn, while a minimum occurs during summer. Note the pronounced inter-annual variability (Fig. S4), which aligns with earlier findings regarding the Foehn timeseries in Altdorf (Richner et al., 2014).

*Deep Foehn* emerges as the most frequent Foehn subtype (89.3%) with a seasonality similar to the overall Foehn timeseries except for a slight frequency shift from spring to autumn. 6.3% of all Foehn hours correspond to *Shallow Foehn*, most of which occur during spring. *Gegenstrom Foehn* is rare (4.4%) and mostly occurs during the winter months, when synoptic situations with a strong zonal pressure gradient over Central Europe develop more frequently. Using four years of station measurements and radiosoundings, Seibert (1990) classified roughly 10% of all Foehn hours in Innsbruck (Austria) as shallow. Considering

**Table 2.** Table providing the Foehn hours for the *Deep Foehn* subtypes, their relative occurrence frequency and seasonality.

| | Dry Foehn | Moist Foehn | Dimmer Foehn |
|---|---|---|---|
| hours | 312 h | 1473 h | 295 h |
| fraction | 13.4% | 63.2% | 12.7% |
| DJF | 11.2% | 20.8% | 19.7% |
| MAM | 54.5% | 45.5% | 46.8% |
| JJA | 10.6% | 4.1% | 3.7% |
| SON | 23.7% | 30.0% | 29.8% |

that they defined *Shallow Foehn* as a Foehn flow with both weak winds or westerlies above crest level, the frequencies for *Shallow Foehn* and *Gegenstrom Foehn* in Altdorf are highly comparable to Innsbruck.

Focusing on the *Deep Foehn* subtypes (Table 2), the occurrence frequencies are evidently given by the percentile approach in the decision tree, which yields a frequency of 13.4% and 12.7% for *Dry Foehn* and *Dimmer Foehn*, respectively. *Dry Foehn* occurs more frequently during spring and summer, in contrast to *Moist Foehn* and *Dimmer Foehn*. To the authors' knowledge, there are, to this point, no studies specifically addressing the seasonality of *Dry Foehn* and *Dimmer Foehn*. Finally, *Moist Foehn* as the main Deep Foehn subtype (63.2%), as well as *Dimmer Foehn* exhibit a seasonality comparable to the overall Foehn timeseries.

Additionally, the overlap of *Dimmer Foehn* and *Gegenstrom Foehn* hours as diagnosed with the decision tree are briefly compared to the corresponding Foehn types determined using station measurements (Section 2.1). In this regard, Table S2 summarizes the most important metrics of the comparison. During the five-year period, the Foehn index identifies both less *Gegenstrom Foehn* and *Dimmer Foehn* hours compared to the classification using the decision tree. First of all, the Foehn index determines these types with the aid of point measurements, which differs from the approach in the study, that considers mean properties within certain regions (Fig. 1b and S3). Secondly, the station-based diagnosis of *Gegenstrom Foehn* applies more rigid thresholds for the wind field (northerly component at Jungfraujoch) and additionally includes relative humidity constraints, a criteria omitted by our classification. In summary, a certain discrepancy between the two classifications is expected. However, when considering a time lag of 6 h, most of the station-based *Gegenstrom Foehn* and *Dimmer Foehn* hours are identically classified with the decision tree (87% and 75%, respectively; see Table S2), which, in fact, provides an observational-based validation for the classification.

## 4 Mean flow conditions during the Foehn types

### 4.1 Synoptic overview

To investigate the mean large-scale conditions during the different main Foehn types and *Deep Foehn* subtypes, composite fields on 500 hPa are calculated using the COSMO-7 analysis data. The main Foehn types are characterised by strikingly different

synoptic conditions (Figs. 3a–c). During *Deep Foehn*, an upper-level trough is situated upstream of Switzerland with its center close to Ireland. Above Altdorf, southwesterlies of 14 m s$^{-1}$ prevail on 500 hPa (Fig. 3a). This flow configuration, usually accompanied by a surface low pressure system over the British Isles and its attendant cold front (not shown), is commonly referred to as the characteristic synoptic situation inducing South Foehn in the Alps (e.g. Richner and Hächler, 2013). It corresponds to the trough Foehn category in Burri et al. (1999), although a pre-frontal Foehn can likewise be classified as *Deep Foehn*, since we do not distinguish between cases with a rapid frontal propagation (e.g., Hächler et al., 2011) and cases with a quasi-stationary cold front (e.g., Burri et al., 1999). In contrast, during *Shallow Foehn*, only a weak, but more elongated upstream upper-level trough is discernible. The Alps are situated in the col of a ridge over the Mediterranean and an anticyclone over Northeast Europe, resulting in very weak synoptic forcing over Central Europe (wind speed of 0.5 m s$^{-1}$ above Altdorf) with a relatively high geopotential on 500 hPa (Fig. 3b). The Alps are approximately situated on the synoptic ridge line. The emerging synoptic situation is comparable to that described in Gerstgrasser (2017). In particular, weak easterlies can be observed over the Adriatic Sea and Northern Italy at 850 hPa (not shown). A similar situation can be observed during the first phase of the case study described in Mayr and Armi (2008), where the ageostrophic cold air advection around the eastern edge of the Alps induces a cross-Alpine gradient in potential temperature and thus causes a shallow Foehn flow. *Gegenstrom Foehn*, in turn, is associated with strong westerlies (25 m s$^{-1}$) and a pronounced geopotential height gradient over Central Europe (zonal flow regime; Fig. 3c). The large-scale situation with a veering of the wind from southwesterlies on 850 hPa to northwesterlies on 300 hPa (not shown) resembles the case study description of Güller (1977), albeit the more northerly flow on the jet level in his original case study.

The subdivision of the *Deep Foehn* subtypes according to the precipitation field reveals that the longitudinal position of the upstream trough is different for each subtype (Figs. 3d–f). During *Dry Foehn*, the amplitude of the upstream geopotential disturbance is reduced and shifted northward. Instead, a ridge emerges over the southern Mediterranean with the synoptic ridge line shifted to the east of Switzerland. Wind speeds are slightly lower (11.6 m s$^{-1}$) compared to *Deep Foehn* conditions. The synoptic situation during *Moist Foehn*, in turn, essentially resembles the *Deep Foehn* composite. This reflects the fact that *Moist Foehn* constitutes the most important *Deep Foehn* subtype. During *Dimmer Foehn*, the trough is deeper and more elongated, which results in a synoptic trough axis closer to the Alps, lower geopotential height above Altdorf and stronger free-tropospheric southwesterlies of 18.5 m s$^{-1}$ on 500 hPa. While the *Deep Foehn* subtypes are distinguishable due to the varying trough position, they can also be envisioned as different stages of the same Foehn event, with a propagation and deepening of the upper-level trough when transitioning in the event's evolution from drier to moister phases.

### 4.2 Mesoscale structure across and along the Alpine crest

North-South cross sections with terrain-following streamlines and upstream precipitation have become a widespread illustration for cross-Alpine Foehn flows in many basic meteorological or high school textbooks (see Seibert, 2005, for a critical review of this development over time). This Section depicts composites of the vertical stratification, moisture distribution and cross-Alpine flow for the different Foehn types using the COSMO-1 analysis. In particular, the composite cross sections can be

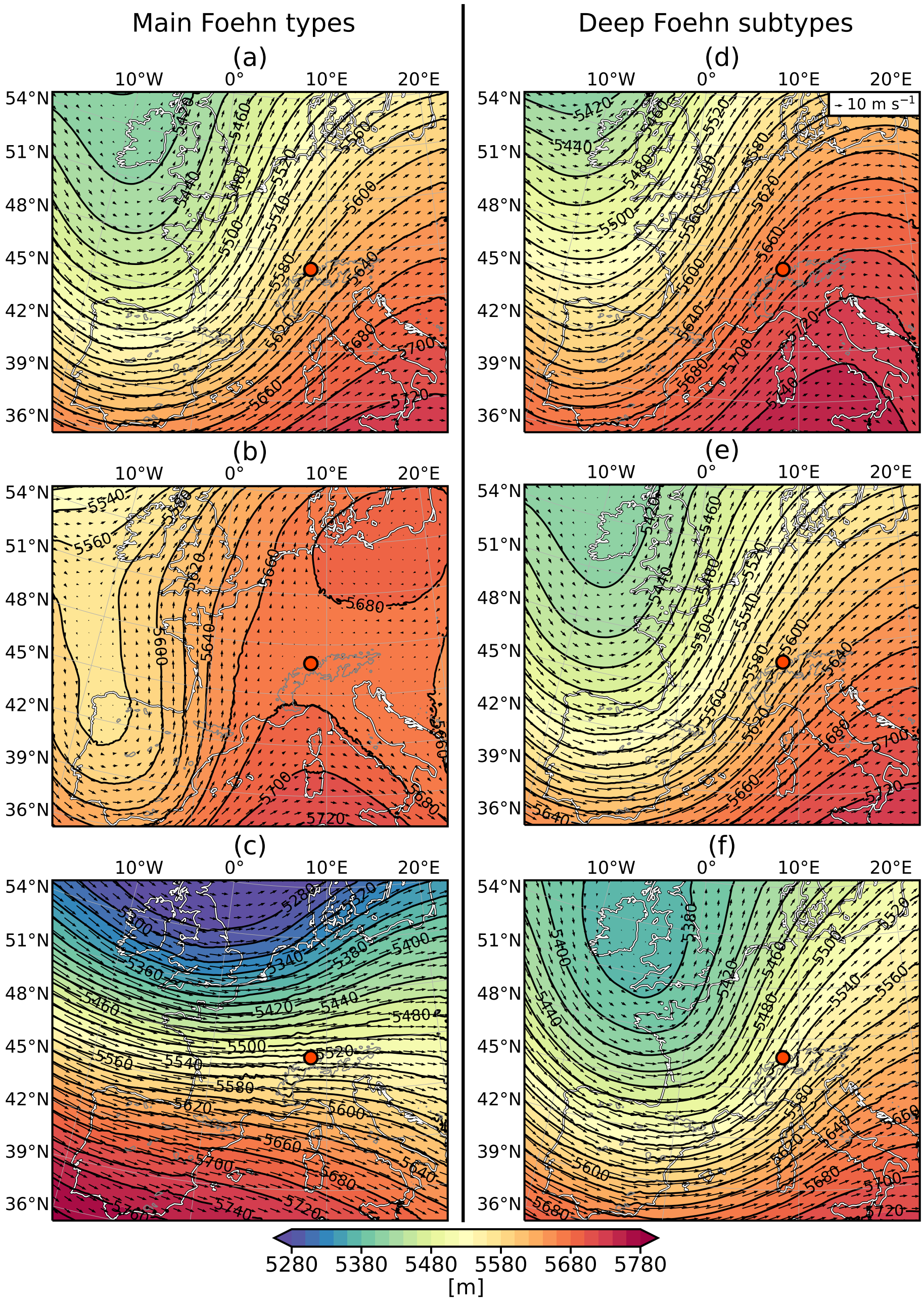

**Figure 3.** COSMO-7 composites of geopotential height (color and black contours) and horizontal wind field (vectors) at 500 hPa for the different main Foehn types and *Deep Foehn* subtypes. (a) *Deep Foehn*; (b) *Shallow Foehn*; (c) *Gegenstrom Foehn*; (d) *Dry Foehn*; (e) *Moist Foehn*; (f) *Dimmer Foehn*. Altdorf is marked as a red dot and the 1500 m contour of the Alps is indicated in grey.

used to see how closely the different Foehn types correspond to the conceptual narrative shown in textbooks. In addition to north-south cross sections (in Fig. 4), west-east cross section composites along the Alpine crest will be shown (in Fig. 5).

During *Deep Foehn*, as discussed in Section 4.1 (Fig. 3a), southerly winds prevail in the mid-troposphere and down to levels below 2 km (not shown). As the southerly flow impinges on the Alps, the air is orographically lifted and, once it reaches saturation, clouds form, which is confirmed by the high relative humidity in the composite on the windward side of the Alpine barrier (Fig. 4a). Likewise, orographic precipitation forms upstream over the Po Valley and the southern side of the Alps with a mean intensity of approximately 1 $mm\,h^{-1}$ (see also Fig. S5a for a precipitation composite). Over the two highest peaks along the cross section, the atmospheric flow gets disturbed by the complex terrain. The deflection of the isentropes[1] indicates vertically propagating gravity waves. The southerly flow peaks in a layer above the crest at about 4 km over the western Swiss Alps and 3 km over the eastern Swiss Alps (Fig. 5a), while the wind speed maxima extend into the gaps of favorably oriented valleys (e.g., the Reuss Valley). In contrast, other valleys do not experience Foehn breakthrough (e.g., part of upper Rhine Valley region)[2]. Considering the isentropes in Fig. 4a as approximate streamlines, air parcels experience a strong descent immediately downstream of the crest associated with the vertically propagating mountain waves. The adiabatic compression leads to a warming of the air parcels and thus a decrease in relative humidity, a typical Foehn effect (see e.g. Richner and Hächler, 2013) clearly emerging along the cross section in the region around Altdorf.

During *Shallow Foehn*, the weak synoptic forcing leads to weaker orographic lifting without noteworthy upwind precipitation (see also Fig. S5b) and no gravity waves emanating from the mountain peaks along the cross section (Fig. 4b). The upstream stratification over the Po Valley differs substantially from the one during *Deep Foehn*: A weakly stratified mixed layer is capped by a strongly stratified layer at 3 km. Above, the free troposphere is relatively dry, as it is typical for large-scale subsidence in anticyclonic weather regimes. However, downslope of the Alpine crest, the Foehn effect is apparent from reduced relative humidity. The along-crest composite (Fig. 5b) highlights the restriction of the Foehn flow to major Alpine south-north transects (gaps), primarily in Central and Eastern Switzerland, whereas calm conditions prevail above crest level. This confirms the findings from MAP describing *Shallow Foehn* along the Austrian Brenner transect as a type of gap flow (Mayr et al., 2007). Focusing again on the cross-Alpine vertical cross section (Fig. 4b), the potential temperature contours point to a temperature gradient across the Alps with colder temperatures on the Alpine south side (note that the same is also true for *Deep Foehn*). The temperature gradient is restricted to below-crest levels ($<$ 3 km) and induces a hydrostatic pressure gradient across the Alps. As a result, *Shallow Foehn* develops as a hydraulic compensation flow driven by the temperature difference of the southern and northern air masses (Mayr and Armi, 2008). Consequently, *Shallow Foehn* can only develop where the gaps along the Alpine crest are low enough for the colder air mass to pass through. Hence, in Switzerland, the high Alps south of the Valais and the Bernese Alps (Fig. S1b) largely inhibit the formation of *Shallow Foehn*, an effect also visible in composites of the horizontal

[1]In order to prevent artificial smoothing of isentropic deflections due to seasonal variations, potential temperature in Fig. 4 is shown with respect to a reference value. This is done by subtracting the mean potential at 3 km altitude of each time step from the actual potential temperature field prior to compositing.

[2]Note that the crestline composite crosses the Valais (Fig. S1b) in its upper part, which, being oriented from the northeast to the southwest, often experiences weak northeasterly winds during Foehn.

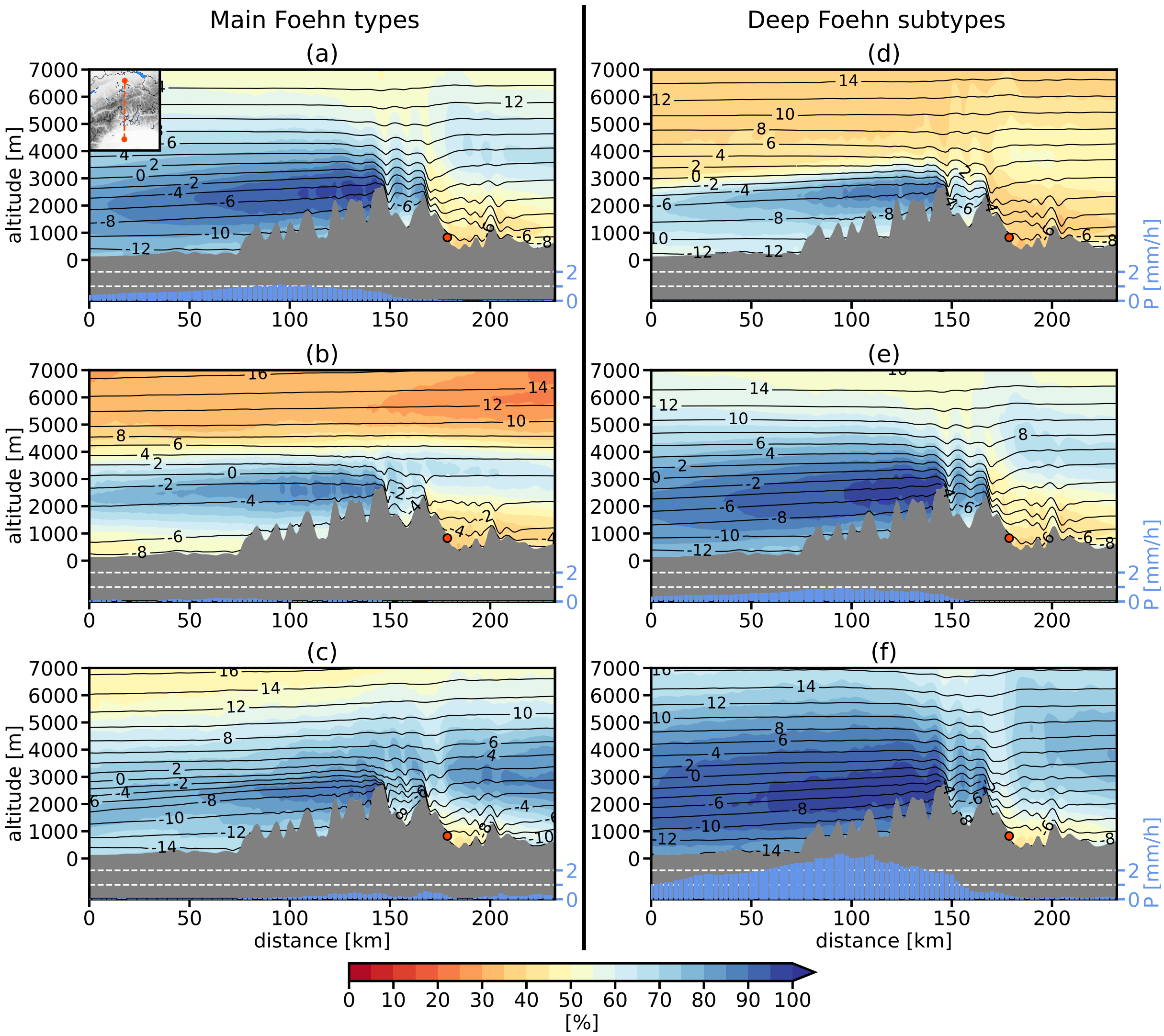

**Figure 4.** Alpine-crossing composites of relative humidity (color) and normalised potential temperature (black contours) in vertical cross sections based on the COSMO-1 analysis. Precipitation along the cross section is indicated as blue bars and the cross section path is displayed as a red line in the inset on the upper left of the first panel. (a) *Deep Foehn*; (b) *Shallow Foehn*; (c) *Gegenstrom Foehn*; (d) *Dry Foehn*; (e) *Moist Foehn*; (f) *Dimmer Foehn*. The latitudinal position closest to Altdorf is marked as a red dot.

wind field on 2.5 km (not shown). This effect will be revisited in Section 6.2 when discussing the influence of the Altdorf Foehn type on the Foehn occurrence frequency at various Foehn locations north of the Alps.

The zonal large-scale flow during *Gegenstrom Foehn* leads to a reduced orographic lifting with weak precipitation over the Alps as well as on the northern side of the Alpine rim (Figs. 4c and S5c). Still, a moderate moistening of air masses south and north of the Alps can be observed. On the Alpine south side, an even more pronounced stable layer, compared to *Shallow*

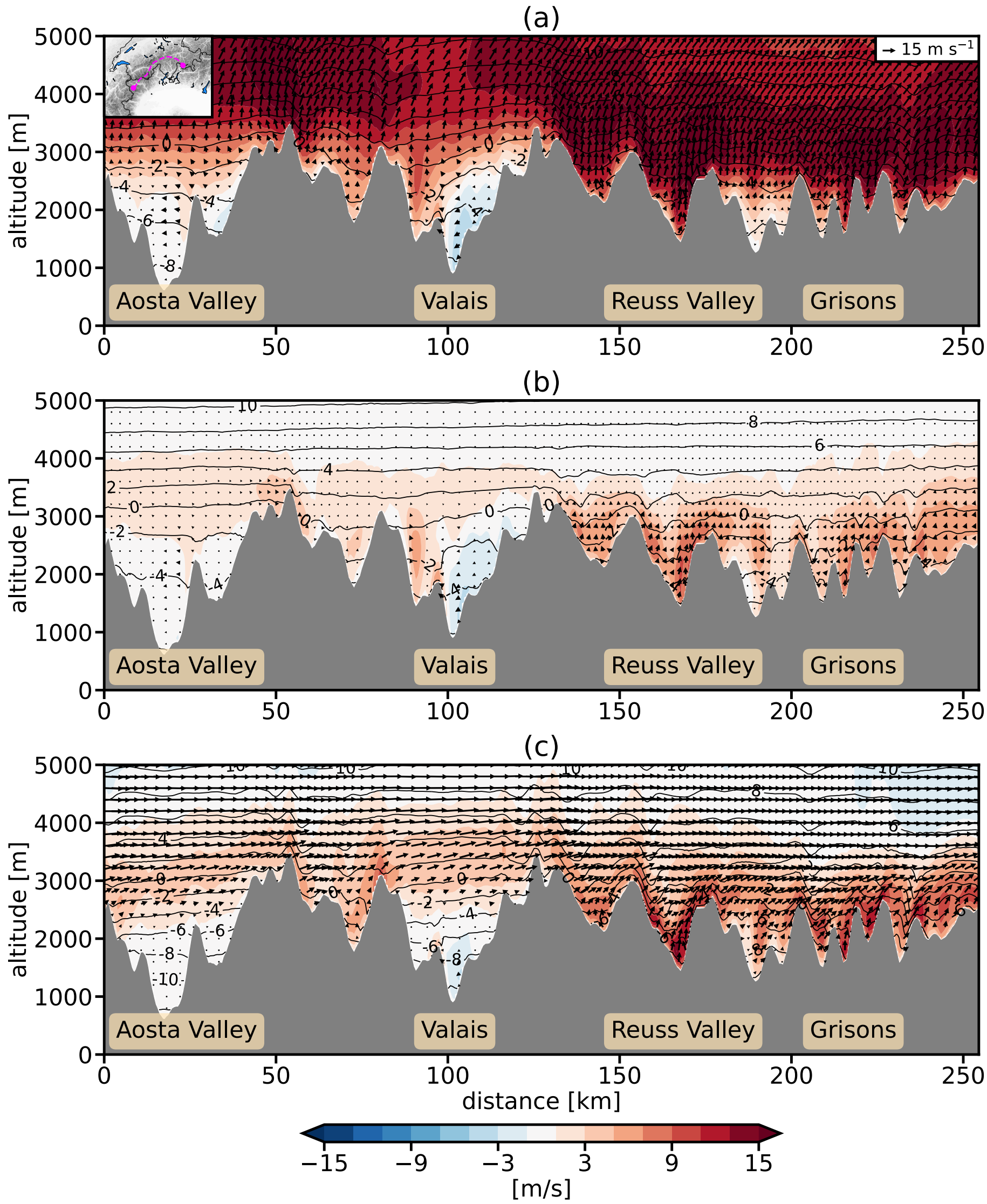

**Figure 5.** Same as Fig. 3 but along the crestline as defined in Section 2.2. The colors indicate the northward component of the wind and the arrows the horizontal wind field. Arrows pointing to the left correspond to eastward winds, arrows pointing upward to northward winds. Major gaps along the crestline are labelled. (a) *Deep Foehn*; (b) *Shallow Foehn*; (c) *Gegenstrom Foehn*.

*Foehn*, is present at 2.5 km altitude. The downslope drying of the Foehn air is less pronounced, possibly owing to the mixing with moist air masses advected with the westerlies north of the Alps. In accordance, Güller (1977) likewise describes the relative humidity as abnormally high during such cases. The Foehn flow is again restricted to levels below 3 km, confining it to major gaps along the Alpine transect (Fig. 5c). Above 3 km, a strong zonal flow prevails. In analogy to the *Shallow Foehn* type, part of the driving cross-Alpine pressure gradient stems from the colder below-crest air mass on the Alpine south side.

However, the synoptic pressure gradient in a zone of enhanced baroclinicity across the Alps (see also the sloping of isentropes above crest level in Fig. 4c) potentially further amplifies the cross-Alpine pressure gradient and thus favors the formation of a Foehn flow.

The differences in depth and elongation of the upstream upper-level trough during the three *Deep Foehn* subtypes evidently has an effect on the vertical distribution of moisture across the Alps (Figs. 4d–f). During *Dry Foehn*, the layer below the capping inversion at 2.5–3 km only reaches moderate saturation levels (relative humidity 60–80%). The reduced upstream moistening also affects the downstream relative humidity values in the Foehn region. Consequently, *Dry Foehn* emerges as the Foehn type with the lowest relative humidity values in the northern Alpine valleys. Again, *Moist Foehn* largely corresponds to the *Deep Foehn* conditions previously described: The southerly flow induces orographic lifting with upstream precipitation and vertically propagating gravity waves. *Dimmer Foehn*, on the other hand, emerges as a Foehn variety accompanied by very intense orographic precipitation ($> 2\,\mathrm{mm\,h^{-1}}$). As opposed to *Moist Foehn*, the atmosphere is not only close to saturation at crest level, but the strong southwesterlies and, potentially, also large-scale lifting downstream of the trough over the British Isles, induce elevated relative humidity values throughout the mid-troposphere. As typical for *Dimmer Foehn*, the strong cross-Alpine flow leads to a partial spillover of precipitation into regions north of the main Alpine crest (note precipitation north of the 150 km mark along the cross section). As a consequence, precipitation reaches the Foehn region around Altdorf (see also Fig. S5f). Accordingly, the relative humidity in the lee-side valley air is higher than during *Moist Foehn*, which is typical for *Dimmer Foehn* according to Richner and Hächler (2013).

In summary, the identified Foehn types with their distinct synoptic situations also differ with respect to their vertical structure (moisture distribution, precipitation, stratification, flow field) along and across the Alps. To some extent, the mean *Deep Foehn* conditions resemble the conceptual textbook model and the characteristics of the 'Swiss Foehn' type (Steinacker, 2006). Nevertheless, the intensity of upstream orographic lifting and precipitation, as well as the presence and strength of a stable layer at crest level, varies considerably for the different Foehn types. As a consequence, one can raise the question how these differences are reflected in the origin and thermodynamic evolution of Foehn air parcels for the different Foehn types. To this aim, the next Section will tackle these questions adopting the Lagrangian perspective.

## 5 Lagrangian analysis

Several recent studies point to the existence and varying importance of different airstreams contributing to Alpine Foehn events (Miltenberger et al., 2016; Jansing and Sprenger, 2022; Saigger and Gohm, 2022). These airstreams differ with respect to the horizontal and vertical origin of Foehn air parcels and the thermodynamic evolution along their transport pathway. This Section will investigate the extensive Foehn data set with respect to their occurrence, air mass origin and the thermodynamic evolution associated with them (Section 5.2). Furthermore, the linkage to the Foehn types and upstream stratification will be discussed (Section 5.3).

### 5.1 Trajectory classification

In order to analyze the occurrence of airstreams during Alpine South Foehn events in a systematic manner and quantify their importance for the different Foehn types, backward trajectories are independently classified for the Foehn types. To this end, the changes of three positional variables (longitude: $\Delta$lon, latitude: $\Delta$lat, altitude: $\Delta$z) and three thermodynamic variables (temperature: $\Delta$T, potential temperature: $\Delta\Theta$, specific humidity: $\Delta$QV) with respect to the Alpine crest are extracted from the compiled trajectory data set (see Section 2.3 for details regarding the trajectory calculations). Specifically, the variables in the set are defined as the difference of the conditions at the Alpine crest relative to the conditions 6 h prior to crest arrival (e.g., $\Delta\text{lon} = \text{lon}_{\text{crest}} - \text{lon}_{\text{upstream}}$). When calculating backward trajectories using a limited-area model, air parcels eventually leave the domain at a certain time after being released. Thus, the reference time of 6 h is chosen as a pragmatic threshold in order to include as many trajectories as possible for further analysis, while, at the same time, capturing the bulk of the upstream ascent for the majority of air parcels. With the chosen threshold, 279'036 trajectories, i.e. 2.8% of all trajectories, are excluded from further analysis since they leave the domain in less than 6 h after having crossed the defined crest line.

For the further analysis, the variable set is standardized and used to perform a principal component analysis (PCA), which serves to reduce the dimensionality of the parameter phase space spanned by the six variables. The resulting biplot (Fig. 6) shows the first and second principal components (PC1 and PC2) and the projections of the original variables into the principal component space (vectors). The plot illustrates that the variability along the first principal component (PC1), explaining most of the variance (60%) in the six-dimensional phase space, is mainly governed by the thermodynamic variables and the altitude difference, which is expected to be strongly correlated to the thermodynamic evolution. Accordingly, variablity along the second principal component (PC2) is largely associated with horizontally varying trajectory positions relative to the crest ($\Delta$lon, $\Delta$lat). Since PC1 explains most of the variance, the trajectory data set is subdivided into terciles with respect to PC1 (see also the histogram with the chosen thresholds indicated as green, dashed lines in Fig. 6). Hence, trajectories with PC1 $\leq$ -0.83 belong to cluster 1, trajectories with -0.83 $<$ PC1 $\leq$ 0.89 to cluster 2, and trajectories with PC1 $>$ 0.89 to cluster 3.

### 5.2 Air mass origin and thermodynamics

As elaborated in the previous section, the trajectory clusters are defined based on PC1, whose variability is mainly governed by thermodynamic variables. However, the three clusters are also associated with a differing upstream origin of air parcels. This also becomes evident from the slope of the point cloud in Fig. 6. Trajectories in cluster 1 predominantly originate in the central to eastern Po Valley (Fig. 7a) at an altitude of 1.5 km or below (Fig. 7b) and thus also ascend considerably while approaching the Alps. The origin of cluster 2 air parcels is shifted more to the west into the region of the Ligurian coast (see Fig. S1a for orientation), as they only experience a slight deflection to the east relative to crest (Fig. 7a). They stem from a mean altitude of 2.5 km, implying a slight ascent towards the crest (Fig. 7b). Finally, trajectories in cluster 3 are subject to the southwesterly flow above crest level and thus their origin is situated over the Maritime Alps (Fig. 7a). They exhibit an additional spread in their longitudinal position compared to cluster 1 and 2, which might be related to a corresponding spread in the Alpine-impinging synoptic flow. During their passage over the Western Alps and the northern Po Valley towards the main crest, they descend

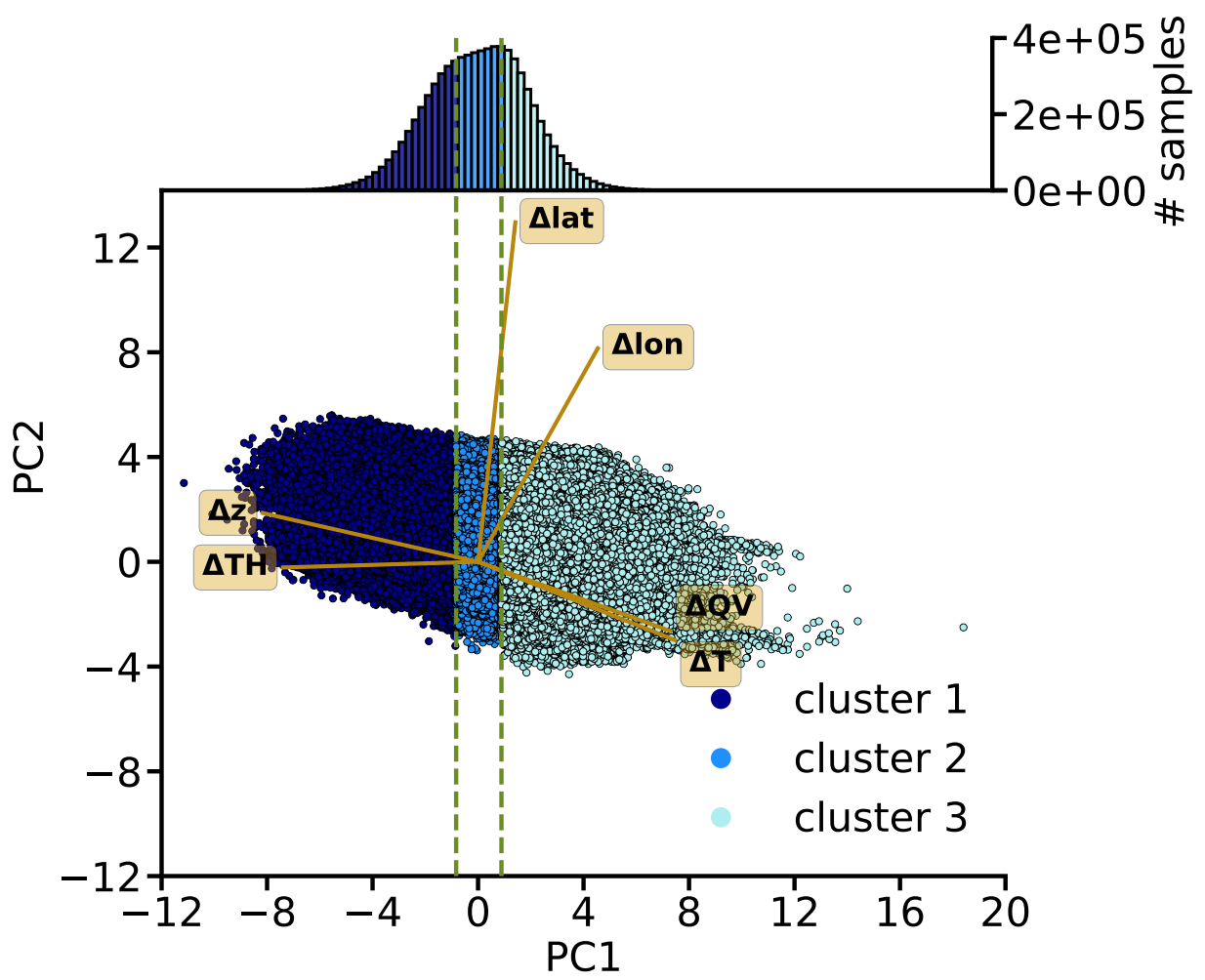

**Figure 6.** Biplot of PC1 and PC2 including the loadings (dark orange lines with labels in light brown boxes). The scores are colored in different shades of blue according to the three clusters based on PC1 terciles. A histogram of PC1 and the 33rd and 66th percentiles (green, dashed lines) are included as an inset above the biplot. For details on the PCA it is referred to the text.

from higher levels (> 3.5 km in the mean; Fig. 7b). Trajectories in all three clusters reach the Alpine crest at a mean altitude slightly lower than (cluster 1 and 2) or at 3 km (cluster 3). The different origin of the three trajectory clusters is linked to the characteristic directional shear pattern upstream of the Alps during Foehn events: Above crest-level, usually a southwesterly to southerly flow prevails (for *Deep Foehn*, see Fig. 3). Below crest, the flow is typically, at least partially, blocked by the Alps. The blocking effect leads to a westward flow deflection and the formation of an easterly barrier jet redirecting air parcels towards the Alpine concavity in the region of Lago Maggiore (Schneidereit and Schär 2000; Medina and Houze 2003; Rotunno and Houze 2007). During heavy precipitation events with an enhanced low-level inflow of moisture, the combined effect of the Coriolis force and the arc-shape of the Alpine barrier, is found to facilitate lifting in the western part of the Po Valley (Schneidereit and Schär, 2000), which potentially explains the considerable ascent of cluster 1 trajectories.

As vertical motion triggers cloud and precipitation formation, the three clusters are also characterised by a different diabatic transport history and varying precipitation intensity. To illustrate this, the thermodynamic evolution of the clusters is examined in T-Θ space (Fig. 8). This diagram is spanned by temperature on the x-axis, and potential temperature on the y-axis. Additionally, isobars are included to show the vertical evolution of the air parcels. Cluster 1 trajectories ascend most strongly (more than 150 hPa) during their approach to the Alpine crest, and therefore experience the most pronounced diabatic heating exceeding 4.5 K in the mean as seen in the Θ-change. At the same time, corresponding to the moist adiabatic lapse rate of ~6 K km$^{-1}$, the moist-adiabatic ascent decreases temperature by 11.5 K. The mean pathway in T-Θ space points to pronounced latent heating during cloud formation. Cluster 2 air parcels are subject to a weak, nearly adiabatic ascent of 25 hPa during the first 3 h. As they approach the Alpine crest, their ascent speed increases. Overall, they experience a moderate diabatic heating of ~1 K. Note that the mean trajectory in T-Θ space of both cluster 1 and 2 reveals that they start descending (increase in temperature

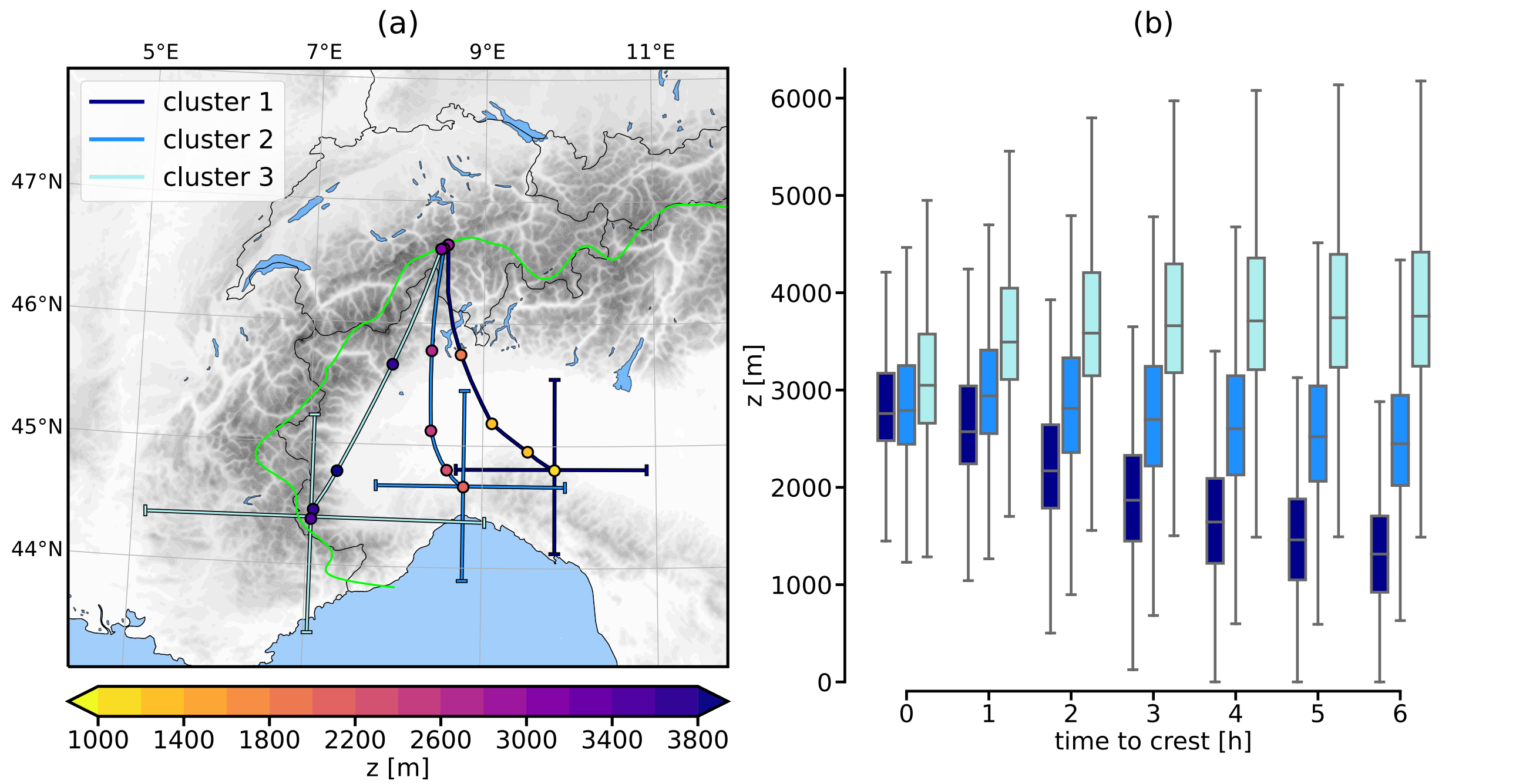

**Figure 7.** (a) Mean pathways of the three trajectory clusters (bold lines in shades of blue) and variability of the position 12 h prior to arrival at the crest (bars ranging from minus to plus one standard deviation in longitudinal and latitudinal direction, respectively). Dots mark every three hours and are colored according to the mean altitude of the clusters at the respective time step. Note that for the data points earlier than 6 h with respect to crest, the mean is calculated by excluding trajectories that left the domain in the meantime. (b) Height distribution of the three trajectory clusters starting from 6 h prior to crest in hourly intervals. The boxplots whiskers range from 1.5 times the interquartile range below the first quartile to 1.5 times above the third quartile, respectively. Outliers are not shown.

and pressure) prior to the crest arrival, defined as intersection with the crestline (lime line in Fig. 7a). This is likely caused by a particularly northern position of the smoothed crestline in the region of Altdorf. Furthermore, the phase line of vertically propagating gravity waves usually tilt upstream (Smith, 1979), which might induce a descent of air parcels even before they have reached the crest. However, the descent appears during a minor segment of the trajectories and, hence, should not effect the main findings.

Cluster 3 is associated with a contrasting thermodynamic evolution compared to the other two clusters. From 6 h until 2 h prior to arrival, the air parcels descend at an almost constant temperature. This indicates that the adiabatic heating during the descent is compensated by an accompanying diabatic cooling of ~1.5 K. As the descent accelerates, mainly during the last hour prior to crest arrival, the trajectories experience a mean warming of ~3.5 K, while they are subject to further diabatic cooling of ~2.5 K. At the crest, cluster 3 trajectories are still at a slightly higher level compared to cluster 1 and 2.

The unique air mass origin and diabatic evolution of the three trajectory clusters can be linked to their upstream humidity changes and surface precipitation (Fig. 9). For this assessment, binned trajectory maps are constructed as follows: All trajectory increments starting from 6 h prior to their arrival are binned onto a regular 0.04° x 0.04° rotated latitude longitude grid (equator

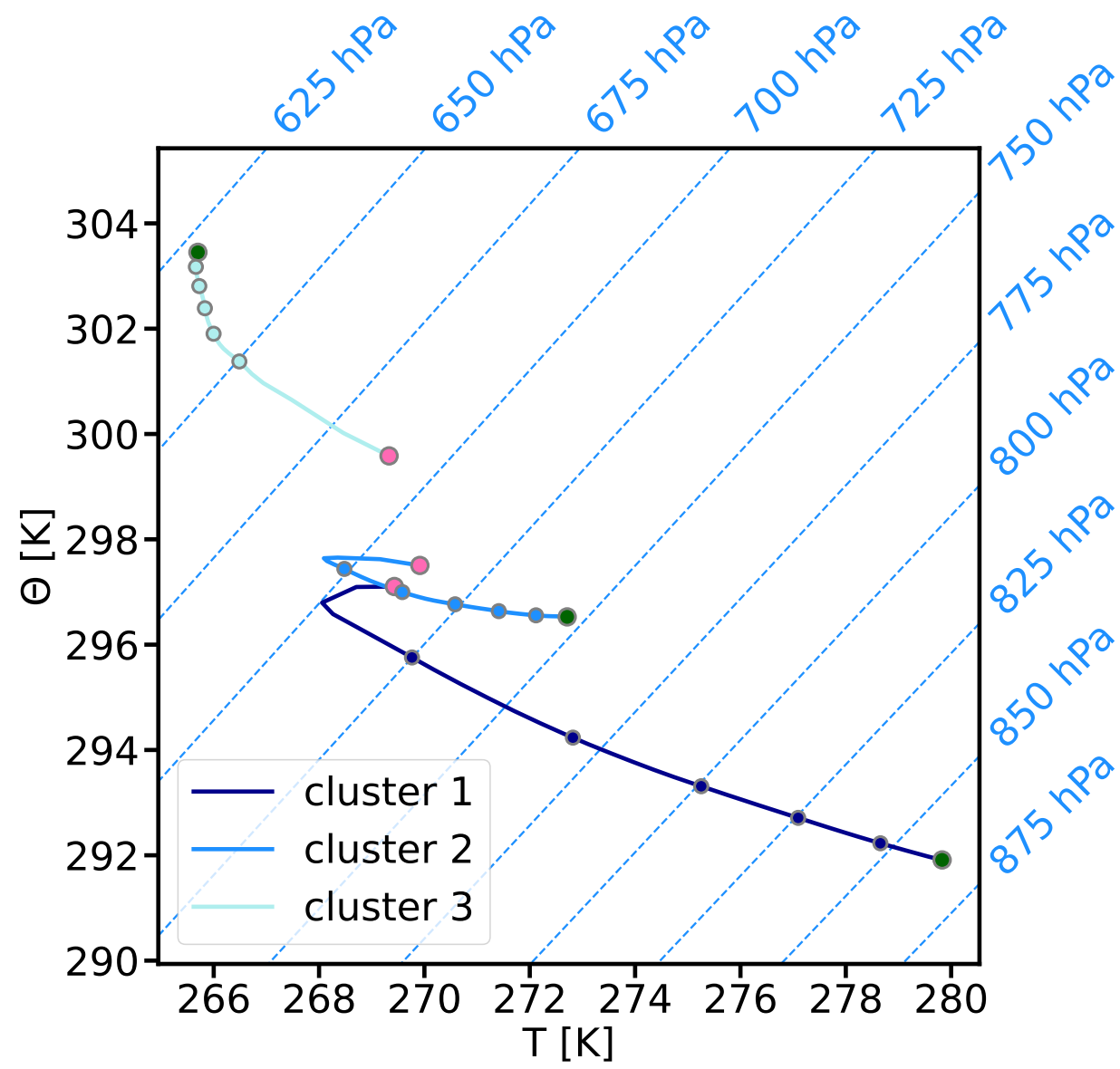

**Figure 8.** Thermodynamic evolution of the mean of each cluster in T-Θ space from 6 h before arrival at crest (green dot) to crest (pink dot). Dots mark each hour and isobars are displayed as blue dashed lines with a spacing of 25 hPa.

at 43° N, zero meridian at 10° E). Within each bin and for each cluster, mean properties are then calculated. Bins with less than 1000 trajectories are masked to enhance visibility and ensure that only representative areas are displayed. Air parcels of cluster 1 lose more than 2.5 $g\,kg^{-1}$ of moisture and are associated with widespread precipitation exceeding 0.5 $mm\,h^{-1}$ (Fig. 9a). Thererfore, cluster 1 emerges as the main precipitating Foehn airstream. Two main regions of strong moisture loss and precipitation can be identifed. Inflow from the central to eastern Po Valley produces precipitation along the southern flanks of the Alps and south of the Lago Maggiore region. Besides, a secondary inflow (less trajectories in cluster 1 originate from this region; see Fig. S6a) from the Mediterranean is related to a pronounced decrease in humidity and precipitation along the Ligurian coast and the Maritime Alps. The humidity loss corroborates that the potential temperature increase is induced by irreversible moist-adiabatic ascent and precipitation formation along the trajectories. Combined with its pronounced diabatic heating and the strong upstream ascent, this cluster corresponds to the classical 'Swiss Foehn' type (Steinacker, 2006; Würsch and Sprenger, 2015).

As opposed to cluster 1, the specific humidity of trajectories from cluster 2 only decreases slightly (−0.5 $g\,kg^{-1}$) and mainly when air parcels originate at the Ligurian coast (see also mean pathway in Fig. 7a). The nearly saturated air mass is confined to the Lago Maggiore area (not shown). Therefore, it can be deduced that cluster 2 is mainly associated with cloud formation and only minor amounts of precipitation.

Finally, and even more so than cluster 2, the trajectories of cluster 3 do not produce precipitation. In contrast, they gain humidity along their pathway towards the crest (0.5 $g\,kg^{-1}$). As they usually travel well above the surface (see median altitude of cluster 3 in Fig. 7b), the moisture gain could either be induced by evaporation of cloud and rain droplets, or result from

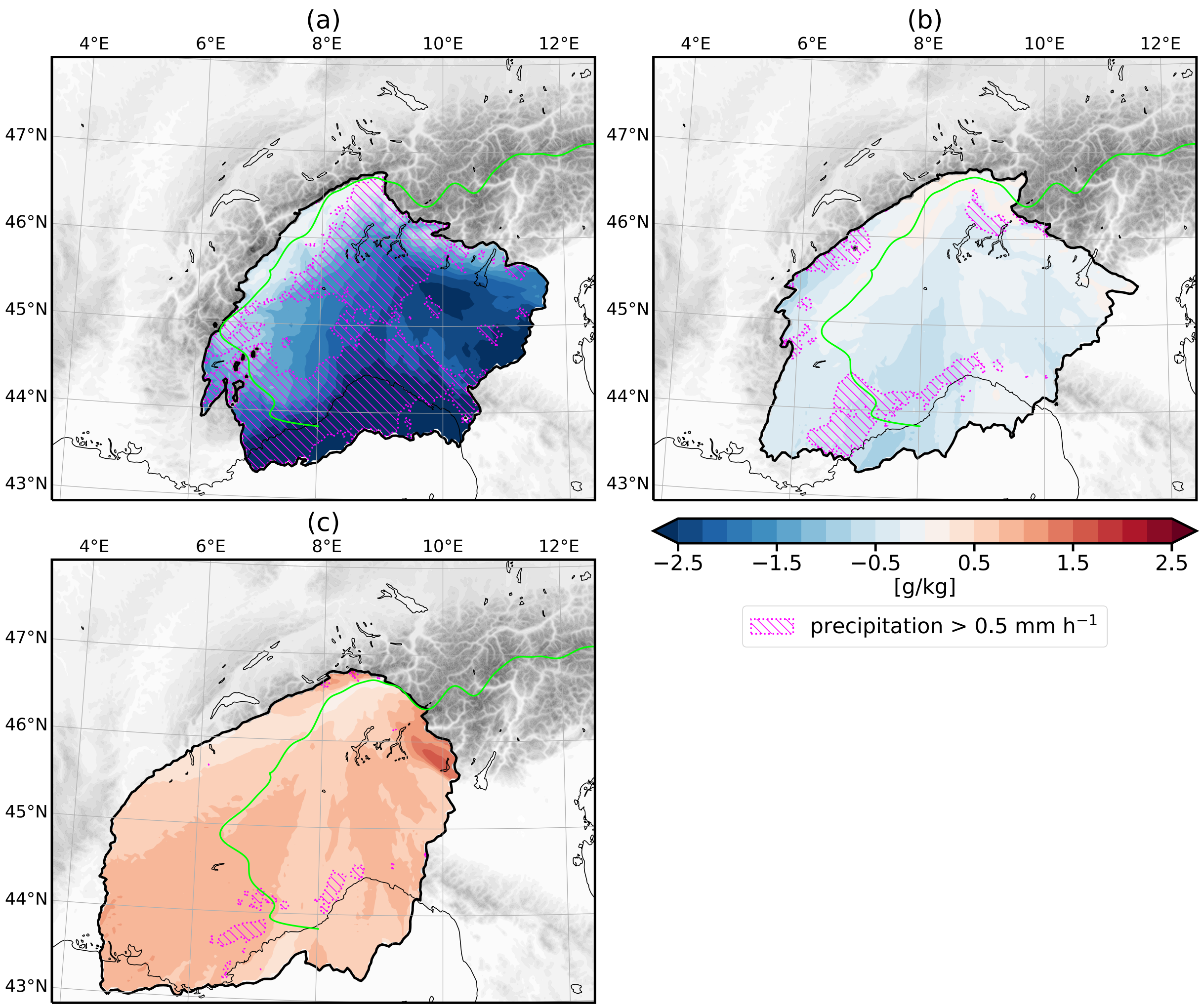

**Figure 9.** Binned trajectory maps with specific humidity changes relative to crest (in color) and precipitation exceeding 0.5 mm h$^{-1}$ (as dashed violet areas) for the different trajectory clusters: (a) cluster 1; (b) cluster 2; (c) cluster 3. Areas with less than 1000 trajectory positions contributing to the mean are masked. See Fig. S6 for density maps depicting the number of trajectories within each bin. The crestline is indicated as a lime line.

entrainment of more humid air from below by vertical turbulent mixing. Likewise, both of these processes (evaporation, turbulent mixing) could be responsible for the diabatic cooling observed along cluster 3 air parcels (Fig. 7b). Since the cloud water content increases as well (not shown), entrainment emerges as the most probable process driving the moisture gain.

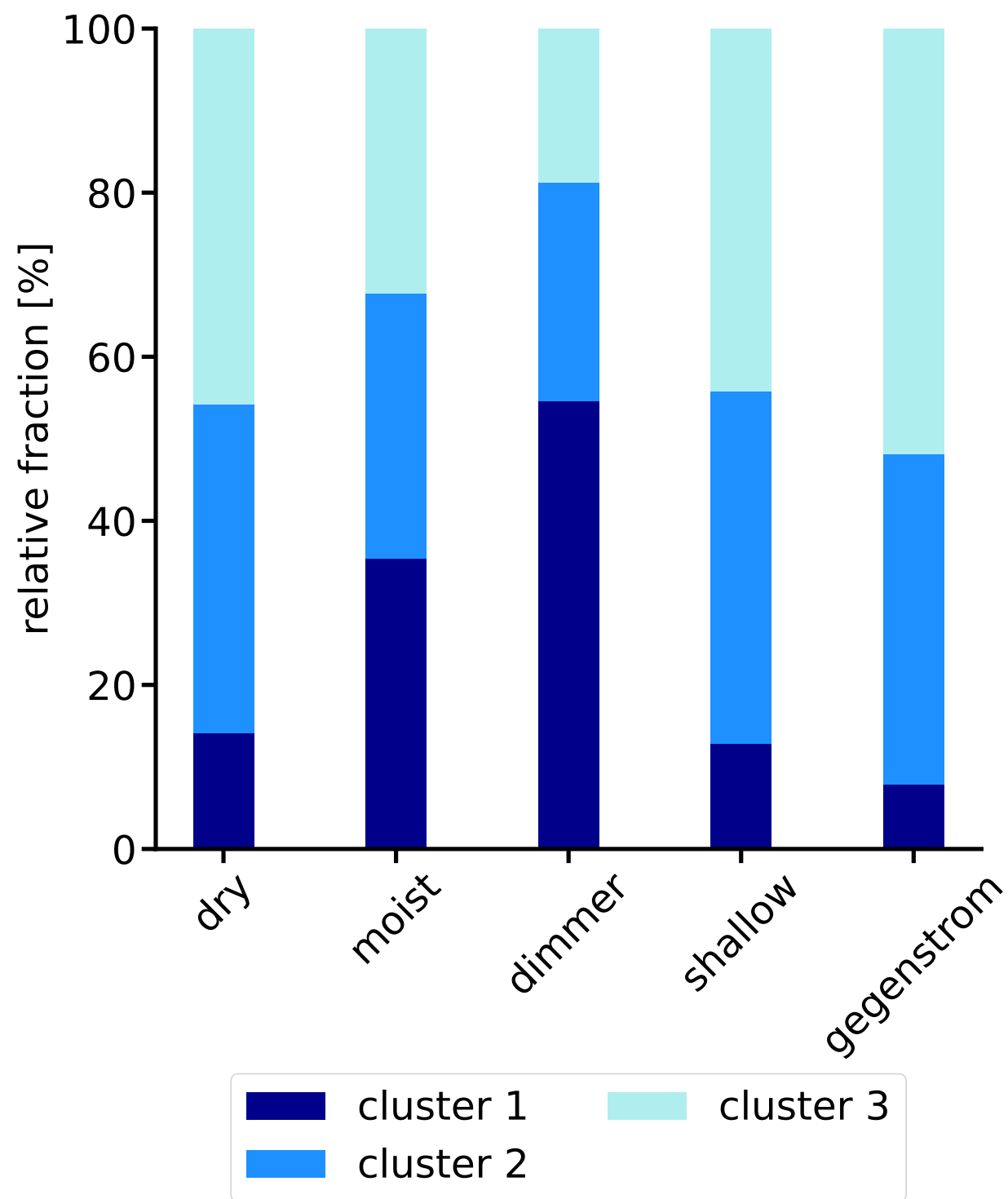

**Figure 10.** Relative contributions of the different trajectory clusters to the overall trajectory sample for the different Foehn types.

### 5.3 Linkage of trajectory clusters to Foehn types

Section 4 revealed substantial differences between the different Foehn types with respect to the intensity of upstream moist processes. Hence, the aim of this Section is to link the Foehn types to the occurrence of the different trajectory clusters that potentially induce these differences. The *Deep Foehn* subtypes will be analysed concurrently with the remaining main Foehn types, since *Deep Foehn* is dominated by the *Moist Foehn* subtype and therefore tends to show the same characteristics as the latter.

At first, it is unravelled how the three trajectory clusters contribute to the different Foehn types. In this regard, the *Deep Foehn* subtypes exhibit particularly pronounced differences (Fig. 10). *Dry Foehn* trajectories mainly belong to clusters 2 and 3, which implies that the air parcels originate from elevated upwind levels and do not experience significant latent heating and no upstream precipitation. A shift in the relative contributions can be observed for *Moist Foehn*, where the contribution of cluster 1 becomes more relevant, resulting in approximately equal contributions of all clusters. This clearly links to the terrain-forced lifting, condensation and orographic precipitation formation as derived from the composite analysis in Section 4.2 (Fig. 4e). For *Dimmer Foehn*, in turn, cluster 1 trajectories are dominant. The latter is in alignment with the Eulerian analysis conducted, where *Dimmer Foehn* emerges as associated with intense and widespread orographic precipitation (Figs. 4f and S5f). *Shallow Foehn* and *Gegenstrom Foehn*, on the other hand, are not related to considerable upstream precipitation (Figs. 4b and c), which

results in a fairly small contribution from cluster 1 trajectories. Cluster 2 and 3 equally contribute to *Shallow Foehn*. Cluster 3, which originates from the southernmost tip of the Alpine arc, constitutes the governing airstream during *Gegenstrom Foehn*, since the large-scale flow above crest level has a pronounced westerly component during *Gegenstrom Foehn*.

The sourcing of Foehn air parcels is linked to the upstream stability as already described by Elvidge et al. (2015) for the Antarctic peninsula. To disentangle this relationship, the vertical stability of the different Foehn types is revisited in combination with the main upstream source altitude of the three trajectory clusters. The zonally oriented cross sections in Fig. 11 are colored according to the mean frequency by which the squared Brunt–Väisälä frequency ($N^2 = g/\Theta * d\Theta/dz$) exceeds $2\times10^{-4}\,s^{-2}$. Altitudes with a higher frequency of exceedance correspond to regions of high static stability and indicate the potential presence of an inversion layer. Besides, the composites also show favourite horizontal and vertical locations where the Foehn air parcels intersect the cross section. To this end, the intersection location of each parcel is extracted and binned along the cross section on a 100 m x 50 m grid in the horizontal and vertical, respectively. The resulting density field is normalized to the total number of trajectory intersections for each Foehn type to allow for a comparison between the different types.

*Dry Foehn* is associated with the frequent presence of a stable layer at 2.5–3.5 km. This stably layer tends to be higher in the Western Alps compared to the Eastern Alps (Fig. 11a). Air parcels, mainly belonging to clusters 2 and 3, either approach the Alps at the top of the stable layer over the Piedmont Alps (cluster 3) or within the layer and at its lower boundary over the eastern Po Valley (cluster 2). The levels below 2 km contribute little to *Dry Foehn* (cluster 1), indicating the potential role of the stable layer that largely inhibits the ascent of air parcels from lower levels. During *Moist Foehn*, elevated frequencies of $N^2$ exceedance occur at the same altitude as for *Dry Foehn*, but the values are significantly reduced (Fig. 11b). As a consequence, the ascent of low-level air parcels is facilitated. The hot-spots of trajectory intersections thus shift from the above-crest level to 2.5 km (cluster 2) and 1.5 km in the eastern to central Po Valley (cluster 1). The stability structure of *Dimmer Foehn*, as indicated by the frequency pattern, essentially corresponds to the one observed during *Moist Foehn* (Fig. 11c). Cluster 1 is dominant during *Dimmer Foehn*, which is why its density is elevated in a widespread region at an altitude of 1.5 km in the central Po Valley. Interestingly, the isentropes indicate a low-level (below 2 km) zonal temperature gradient in the Po Valley during both *Moist Foehn* and *Dimmer Foehn* (Figs. 11b and c). As cluster 1 is associated with the easterlies of the Po Valley barrier jet (see also Section 5.2), isentropic upglide might additionally support the low-level ascent of air parcels during moist Foehn events.

*Shallow Foehn* is associated with a stable layer above crest level (3.5–4.5 km) in the western Po Valley and extending over the Western Alps (Fig. 11d). Accordingly, the majority of cluster 1 air parcels approach the Alps from above 3 km and, thus, in the lower part of the stable layer. Cluster 2 trajectories, on the other hand, intersect the cross section well below the stable layer. In fact, some air parcels even originate from near-surface levels. The lowermost part of the atmosphere over the Po Valley is weakly stratified, as already described in Section 4.2 (Fig. 4b). The weak stratification indicates that the Po Valley boundary layer is usually well-mixed during *Shallow Foehn*. As a result, the boundary layer mixing feeds some air parcel into the main airflow crossing the Alps through major gaps. The downward extent of the trajectory densities illustrates that a first-order estimation of the altitude of origin, purely based on the presence of a stable layer, might be misleading.

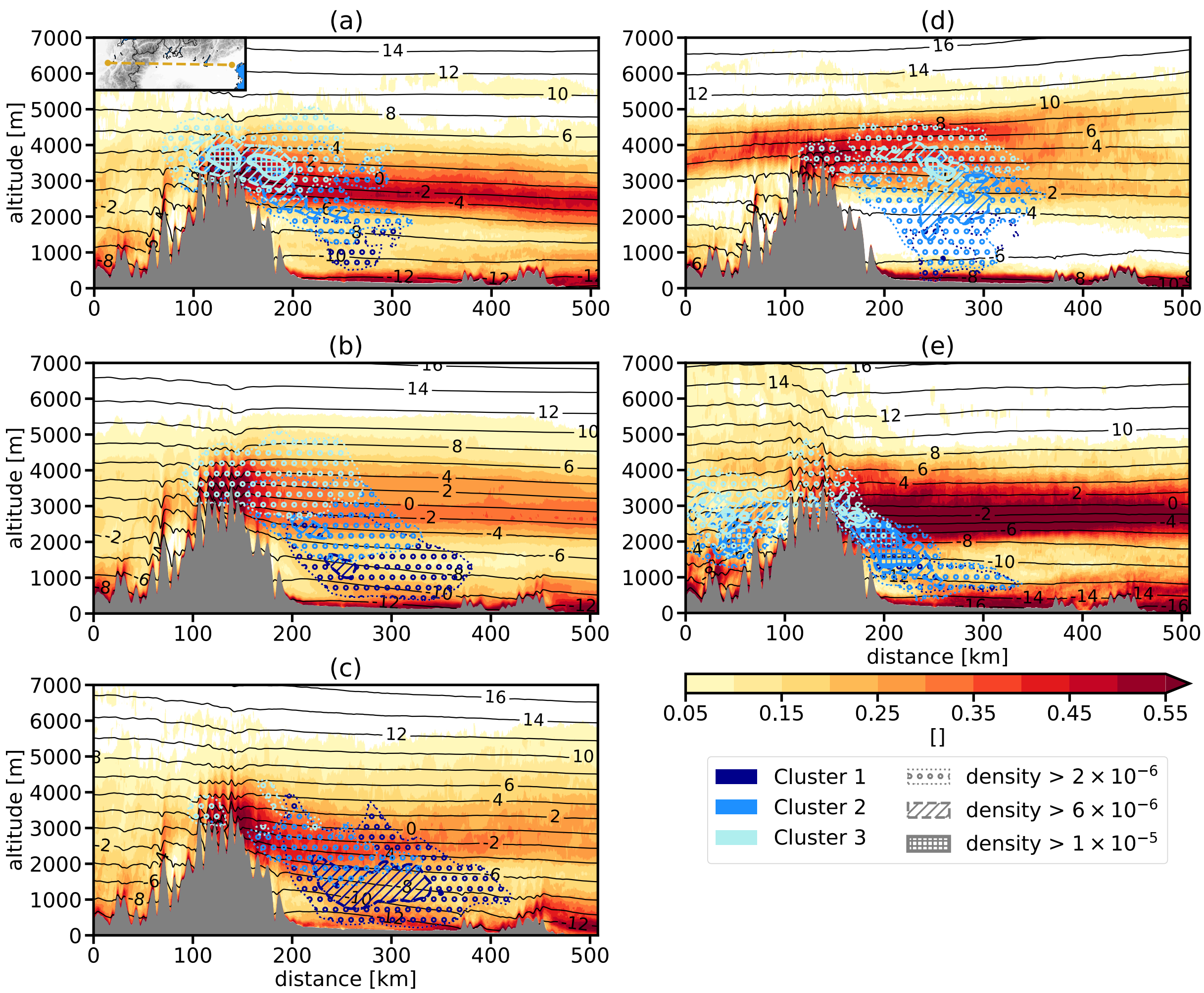

**Figure 11.** Frequency of $N^2$ exceeding $2{\times}10^{-4}$ s$^{-2}$ in vertical cross section composites. Trajectory intersections for the three clusters (different shades of blue) are binned along the cross section and displayed as hatched contours (density $> 2\times10^{-6}$ with circles, $> 6\times10^{-6}$ with lines, $> 1\times10^{-5}$ with squares). Normalised potential temperature is displayed (black contours) as in Fig. 4. (a) *Dry Foehn*; (b) *Moist Foehn*; (c) *Dimmer Foehn*; (d) *Shallow Foehn*; (e) *Gegenstrom Foehn*.

*Gegenstrom Foehn* is characterised by the most pronounced stable layer of all Foehn types (Fig. 11e), which extends over the entire Po Valley in an altitude range between 2.5–3.5 km. In contrast to the other Foehn types, a considerable fraction of cluster 3 and cluster 2 trajectories approach the Alps from the most southwestern tip of the Alpine arc. The strong southwesterlies prevailing during *Gegenstrom Foehn* might support the lifting of these air parcels along the western flank of the Alps. A second hot-spot of trajectories can be found in the Piedmont Alps along the eastern flank of the Alpine arc. Trajectories originate at levels down to below 1.5 km despite the pronounced stable layer above. Either, these air parcels reach just the lower limit of the stable layer and cross the Alps through the major north-south gaps (see Fig. 5), or they are subject to pronounced mechanical

lifting despite the strong stratification. Furthermore, the northward slope of isentropes during *Gegenstrom Foehn* (Fig. 4c) support adiabatic ascent as air parcels approach the crest.

In summary, the Lagrangian analysis reveals a different air mass origin and altitude for each of the thermodynamically defined trajectory clusters. The air parcels originating more to the east generally ascend from lower levels, while the trajectories from above crest-level are advected with the southwesterly large-scale flow. The diabatic heating is more pronounced for trajectories associated with stronger upstream ascent, which confirms the presence of latent heating in clouds for these air parcels. Cluster 1 trajectories are associated with a pronounced moisture loss and widespread precipitation. Hence, this cluster is mainly relevant during *Moist Foehn* and *Dimmer Foehn*, while cluster 2 and 3 are more important for *Dry Foehn*, *Shallow Foehn* and *Gegenstrom Foehn*. The upstream stratification is found to partially govern the altitude from where air parcels approach the Alps. However, additional processes play a role as well (e.g., turbulent mixing) and it becomes clear that the strength of the upstream stable layer alone cannot be considered an unambiguous predictor of the source altitude for Foehn air parcels.

## 6 Impact of Foehn types in Altdorf and beyond

### 6.1 Local conditions in Altdorf

The previous Sections 4 and 5 focused on an in-depth investigation of Eulerian and Lagrangian characteristics of the different Alpine South Foehn types on the mesoscale. This last results Section complements the analysis by examining the local meteorological impact of the Foehn types at Altdorf in the Reuss Valley (see red dot in Fig. 1b). This is of special interest to forecasters, who aim at an accurate prediction of local surface conditions, e.g., in Foehn valleys. In this context, it is valuable to know the characteristic conditions at Altdorf related to a certain Foehn type as diagnosed by the synoptic conditions. To this end, station measurements at Altorf are grouped according to the prevailing Foehn type (Fig. 12).

The *Deep Foehn* subtypes do not exhibit significant differences with respect to the observed wind speed and gusts (Figs. 12a and b). All the subtypes are associated with mean winds ranging from approximately 10–65 km $h^{-1}$ and a median slightly below 40 km $h^{-1}$. Likewise, the gusts range from 15–105 km $h^{-1}$ and exhibit a median of ~60 km $h^{-1}$. The degree of similarity with respect to the flow conditions is, on the one hand, surprising, as the large-scale flow gets continuously stronger for the moister subtypes (*Moist Foehn* and *Dimmer Foehn*) compared to *Dry Foehn*. In agreement, the cross-Alpine pressure difference between Lugano (a station on the Alpine south side) and Kloten (on the Alpine north side) tends be largest for *Dimmer Foehn* and smallest for *Dry Foehn* (see Fig. S7 in Supplement depicting pressure differences between Lugano and Kloten for the different Foehn types). However, all of the *Deep Foehn* subtypes, are prone to gravity wave activity, which might constrain the observed wind speeds.

*Shallow Foehn* emerges as considerably weaker (mean winds of 10–30 km $h^{-1}$, gusts of 20–55 km $h^{-1}$) compared to the other Foehn types. This matches with the along-crest composites (Fig. 5b), where *Shallow Foehn* also manifests as the Foehn type with the weakest cross-Alpine flow. During *Shallow Foehn*, the cross-Alpine pressure gradient is primarily of hydrostatic nature

due to the colder air mass on the Alpine south side. Hence, it is less pronounced than during the other Foehn types (Fig. S7). Potentially, the absence of gravity wave activity (see Section 4.2) acts to further reduce wind speeds within the Foehn valley.

*Gegenstrom Foehn* exhibits slightly reduced mean winds and gusts compared to the *Deep Foehn* subtpyes, but is, at the same time, considerably stronger compared to *Shallow Foehn*. Mean wind speeds range from 5 km $h^{-1}$ up to 65 km $h^{-1}$ with a median of 32 km $h^{-1}$, while the gusts range from 20–100 km $h^{-1}$. The stronger winds and gusts can, again, be explained considering the cross-Alpine pressure gradient (Fig. S7). During *Gegenstrom Foehn*, the gradient is of comparable magnitude as for the *Deep Foehn* subtypes with the overall largest median difference (~8 hPa). The pressure gradient during *Gegenstrom Foehn* can be additionally amplified owing to two reasons. First of all, a pronounced synoptic pressure gradient is superimposed to the hydrostatic pressure gradient generated by the low-level temperature differences (see Fig. 3c). Secondly, *Gegenstrom Foehn* events are oftentimes characterised by the passage of secondary storm lows, e.g., over northern Germany (Güller, 1977; Gerstgrasser, 2017). When such disturbances approach Central Europe, they induce a rapid pressure drop on the Alpine north side. Potentially, this leads to the stronger, more gusty characteristic of *Gegenstrom Foehn* in comparison to *Shallow Foehn*.

The dryness of Foehn winds is heavily invoked as one of their archetypcal characteristics (e.g., Richner and Hächler, 2013). Considering the relative humidity of the different Foehn types, however, a pronounced variability can be observed (Fig. 12c). The overall lowest values in relative humidity are reached for *Dry Foehn* (17–40 %; median at 30%). The large-scale subsidence during *Dry Foehn* inhibits precipitation, and sometimes even cloud formation upwind of the Alpine range (Gerstgrasser, 2017). The free troposphere above the stable layer at crest level, constituting the main source region for air parcels of this *Deep Foehn* subtype (Fig. 11a), is typically relatively dry (Fig. 4d). As a consequence, unusually low relative humidity values can be observed in the lee-side Foehn valleys. The observed relative humidity values tend to be larger for the moister *Deep Foehn* subtypes. The median of *Moist Foehn* and *Dimmer Foehn* amount to 34% and 39%, respectively. Strikingly, the distribution of *Dimmer Foehn* is heavily long-tailed towards higher values. The partial spillover of precipitation onto the leeward side increases relative humidity values of the precipitating air in cases when it reaches Altdorf. Elevated relative humidity values are a typical characteristic of *Dimmer Foehn* (Richner and Hächler, 2013). *Shallow Foehn* is associated with similar relative humidity values as *Moist Foehn*, ranging from 20 to 45%. *Gegenstrom Foehn*, on the other hand, likewise tends to be associated with elevated relative humidity. In contrast to *Dimmer Foehn*, the moister air mass does not result from a partial spillover over the crest, but rather from the mixing of the drier Foehn air with the moist zonal flow north of the Alps (see also Fig. S5c for the precipitation composite of *Gegenstrom Foehn*).

Additionally, the temperature increase at the station owing to the Foehn breakthrough is compared for the different Foehn types (Fig. 12d). It is quantified by subtracting the temperature 3 h prior to Foehn onset from the instantaneous temperature during Foehn. To define the time of Foehn onset, the 2329 Foehn hours are grouped into continuous Foehn events, where an event is defined as a consecutive occurrence of Foehn hours with a maximum Foehn pause of 3 h (resulting in 168 events). This simple approach neglects the influence of the daily cycle on temperature differences, which is, however, strongly dampened during Foehn in Altdorf. For an example timeseries of temperature in Altdorf during Foehn, the reader is referred to Richner and Hächler (2013). The warming is strongest for *Dry Foehn* and *Moist Foehn*, for which the median amounts to 7.5 K and 8 K, respectively. The temperature differences exhibit a considerable spread between below-zero values to values exceeding

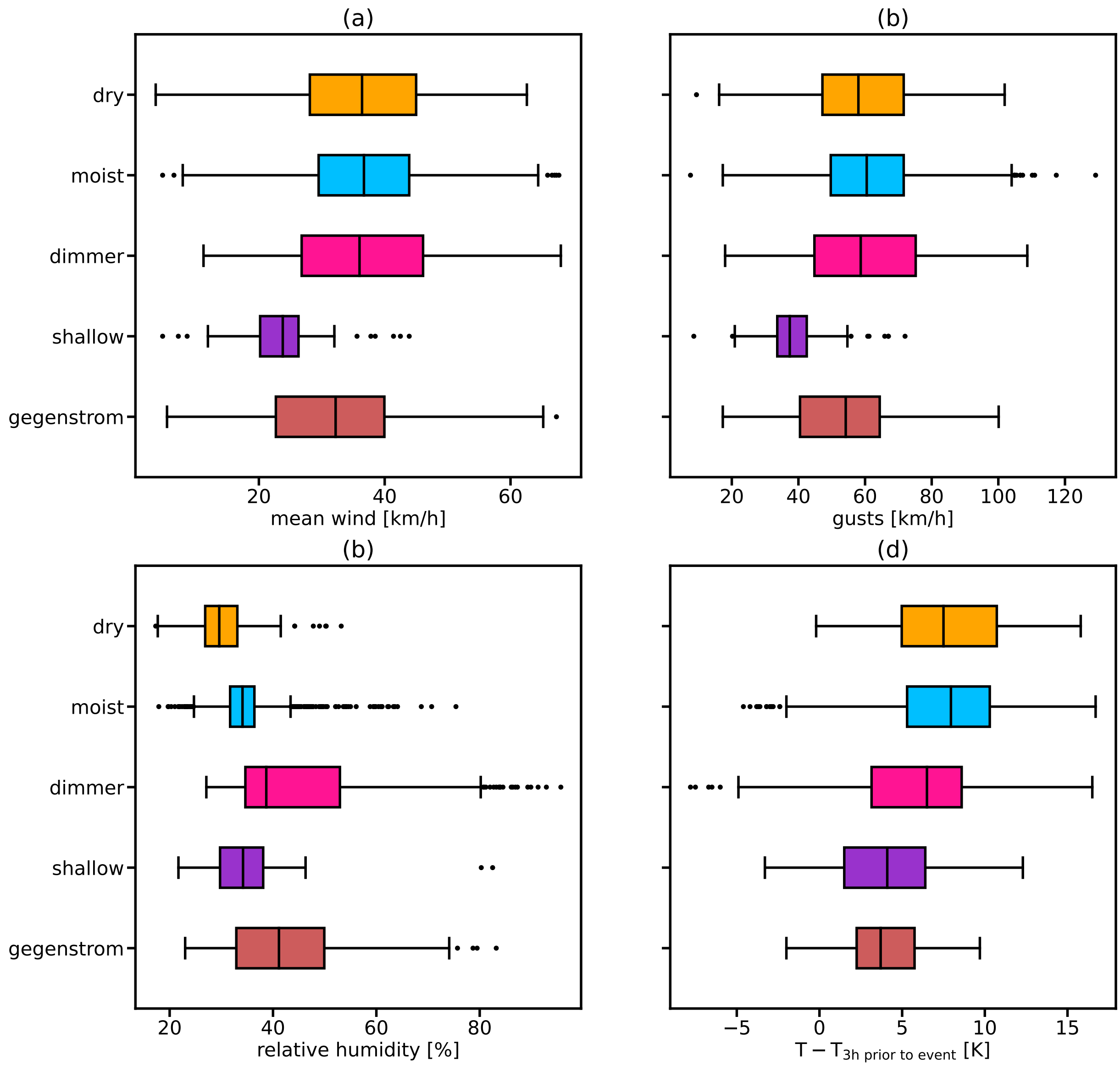

**Figure 12.** Boxplots of Altdorf station measurements of (a) mean wind speed, (b) wind gusts, (c) relative humidity and (d) the temperature difference for the different Foehn types.

15 K. Outliers on both sides of the distribution for the different Foehn types are most probably related to either an influence of the daily cycle, or a rapid air mass change during Foehn breakdown. *Dimmer Foehn* is associated with a less pronounced temperature increase than the other *Deep Foehn* subtypes (median of 6.5 K). This can be explained by considering the thermodynamic history of *Dimmer Foehn* air parcels. As precipitation falls into the sub-saturated atmosphere of the Foehn valleys, rain evaporation induces lee-side diabatic cooling. The cooling counteracts the adiabatic warming induced by the descent and

leads to an overall reduced warming. The magnitude of the warming is further lowered during *Shallow Foehn* (~4 K). A potential explanation considers the gap flow characteristics of such events. As air parcels only cross the Alpine crest through major gaps, they are not able to gain as much elevation difference as if descending from crest level during *Deep Foehn*, hence the Foehn air warming due to isentropic drawdown is reduced. However, systematically assessing the penetration pathways of *Shallow Foehn* air parcels is beyond the scope of this study and could only be accomplished using online trajectories (e.g., Jansing and Sprenger, 2022) or offline trajectories calculated with a much higher temporal input frequency of the 3D wind fields (e.g., Saigger and Gohm, 2022). Finally, *Gegenstrom Foehn* is also characterised by a reduced warming of ~4 K. Similar potential mechanisms might reduce the warming effect, again including a reduced descent of air parcels or evaporative cooling downstream of the Alpine crest. Güller (1977) already quantifies the temperature difference of the Foehn air and the non-Foehn westerlies to be 4 K during his case study. *Gegenstrom Foehn* is also known as a rather 'cold' Foehn type among forecasters. Further, more process-based investigations would be beneficial to improve the understanding of these differences in the Foehn air warming for the different Foehn types.

Overall, the investigation of the local-scale impact revealed pronounced differences between the different Foehn types. The *Deep Foehn* subtypes and *Gegenstrom Foehn* are associated with stronger winds compared to *Shallow Foehn*. This is presumably, at least partly, governed by differences in the cross-Alpine pressure gradient. The characteristic dryness of the Foehn wind at Altdorf varies, too. Relative humidity values are likely to be influenced by the recent thermodynamic history of the air arriving within at the valley floor. The same is also true for the temperature increase at the station. In cases, where precipitation is advected over the Alpine crest (*Dimmer Foehn*) or with the upper-level westerlies (*Gegenstrom Foehn*), below-cloud rain evaporation acts to cool and moisten the Foehn air.

### 6.2 Spatial variability conditional on Altdorf Foehn

The Foehn classification is entirely based upon the timeseries of Foehn occurrence at Altdorf (Reuss Valley). Yet, the Alpine South Foehn occurs in a multitude of different valleys north of the crest and, seldomly, further expands into the Swiss Plateau (see Jansing and Sprenger, 2022, for an example of five typical Swiss Foehn valleys). In a last step, this Section considers the spatial variability of the Foehn types in Switzerland. Since the COSMO analyses are only extracted based on the Foehn observations at Altdorf, we cannot perform the classification according to the decision tree (Fig. 2) for other Foehn locations. The investigation is thus done indirectly via quantifying the Foehn frequency at various stations conditioned on Foehn of a certain type in Altdorf. While this approach cannot explicitly categorize Foehn observations at the other stations, it allows for a relative comparison. This should be interpreted as a first guess whether a location is more or less prone to the occurrence of a certain Foehn type. As the Foehn types are related to distinct synoptic patterns (Fig. 3), it is likely that most Foehn events co-occurring at several stations are of the same type (with the exception of *Dimmer Foehn*, which is a very local phenomenon).

The Foehn frequency conditioned on the occurrence of a certain Foehn type in Altdorf is depicted in Fig. 13. A list of all the stations, including their mean Foehn frequency over the five-year time period and the frequency relative to Altdorf is included in the Supplement (Table S1). Generally, Foehn occurs more frequently closer to the Alpine crest (e.g., Visp) and at higher-altitude stations (e.g., Andeer). Foehn breakthrough gets more likely when the height difference to the crest (or the lowest

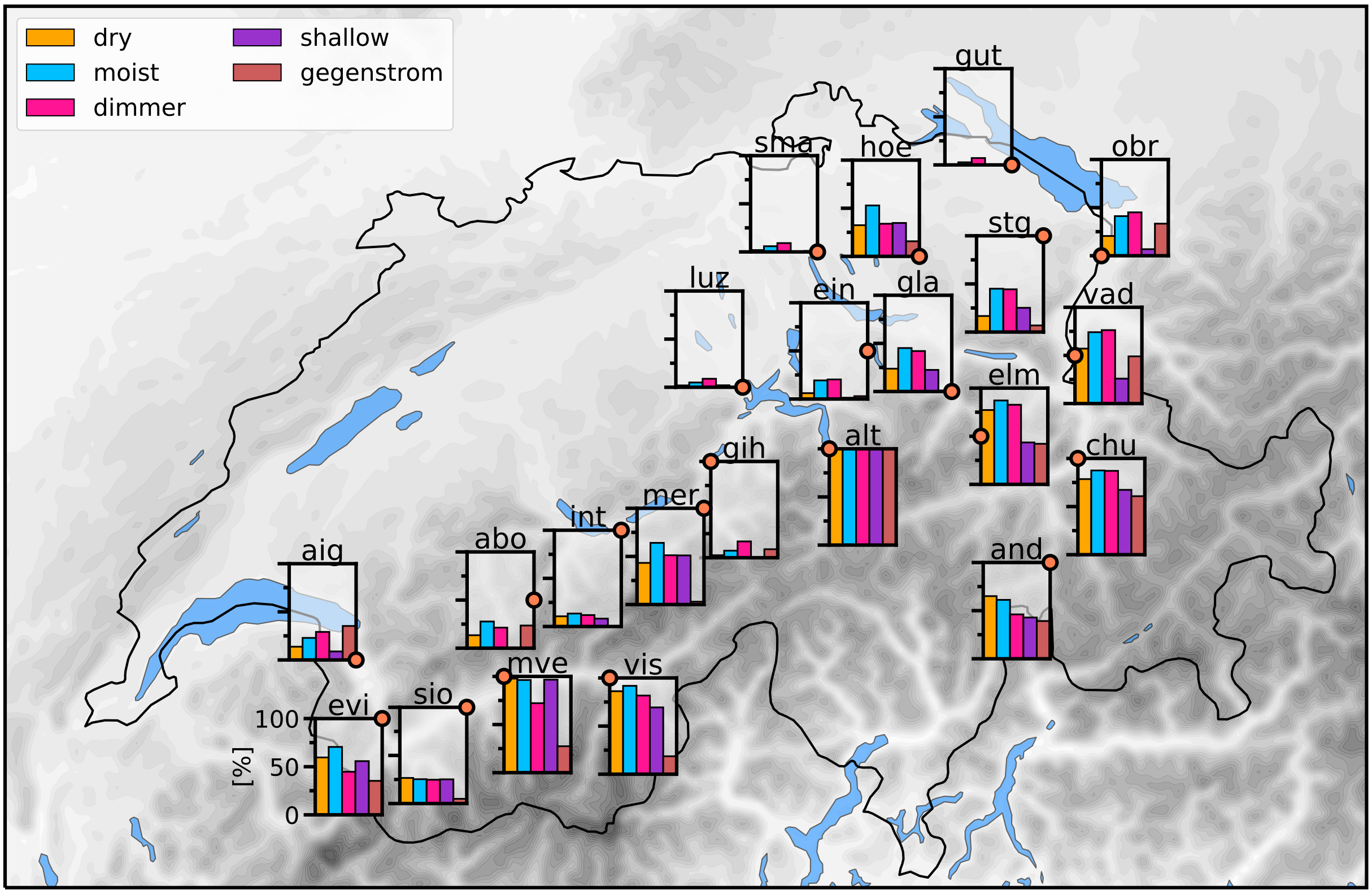

**Figure 13.** Occurrence frequency of Foehn at various stations in Switzerland conditional to the occurrence of a certain Foehn type in Altdorf. The COSMO-1 topography is included in the background for orientation. A detailed list of the stations can be found in the Supplement (Table S1).

gap) upstream is smaller and the descending Foehn flow needs to cover less distance from the crest to the station. Gutermann et al. (2012) already identified the same relationship using observations over a longer time period (1984–2008). To further corroborate this finding, the same is observed when considering multiple stations in a down-valley sequence (e.g., compare Evionnaz with Aigle, Vaduz with Oberriet, or Elm with Glarus). Besides, the frequent presence of cold-air pools during the cold season, covering the Swiss Plateau and extending into the valleys, oftentimes prevents the Foehn to break through down to the station level at locations further down-valley. In addition, local effects certainly modulate the overall Foehn frequency at the different stations. Among them are the local orientation of the valley axes or the interaction of the Foehn flow with diurnal valley winds (see also in Gutermann et al., 2012).

The down-valley frequency decrease is especially pronounced during *Shallow Foehn* in Altdorf. Accordingly, Aigle or Oberriet rarely observe Foehn during such events. *Shallow Foehn* has been identified as the weakest Foehn flow. Thus, down-valley penetration is further hindered. Additionally, the upstream crest height likely plays a role for the occurrence of *Shallow Foehn* as well. Adelboden in the Bernese Alps does not observe any Foehn when *Shallow Foehn* occurs in Altdorf. This is related to the driving mechanism of this Foehn type, which is the colder low-level air mass on the Alpine south side. If this air

mass does not reach crest or gap level, *Shallow Foehn* cannot develop, a situation that is more likely for regions with higher crest as it is the case in the Bernese Alps.

Another striking feature is the pronounced reduction in the frequency of *Gegenstrom Foehn* in the Central Valais (Sion, Montana, Visp). One of the potential explanations for this finding, which is confirmed by forecaster's everyday experience, invokes the pronounced westerlies above crest level. As a consequence, the large-scale flow acts exactly opposite to the Foehn flow and the cold air pool within the Valais cannot be displaced by the Foehn winds. Furthermore, the Valais and the Western Alps more often experience precipitation during *Gegenstrom Foehn* compared to the Rhine Valley (Fig. S5c). Precipitation cools and stabilizes the valley atmosphere, which can inhibit the penetration of the Foehn flow. Similar mechanisms might, oftentimes, prevent the breakthrough of *Gegenstrom Foehn* in Meiringen, as the Hasli Valley is southeast-northwest oriented. In addition, stations that are not well shielded by topographical features to the east are less prone to *Gegenstrom Foehn*, as the near-surface southwesterlies over the Swiss Plateau make Foehn breakthrough more difficult. Among them are, e.g., St. Gallen and Hoernli.

*Dimmer Foehn* also exhibits a distinct conditional frequency distribution. Stations close to the main Alpine crest (e.g., Visp, Andeer) tend to have slightly reduced occurrence frequencies compared to *Moist Foehn*. With an increasing distance to the crest, this relationship however reverses. In particular, rare Foehn events that extend onto the Swiss Plateau occur more often during *Dimmer Foehn* in Altorf (Giswil, Luzern, Zurich, Guettingen). A characteristic of *Dimmer Foehn* is the unusual extent of the Foehn wall beyond the main crest and into the Foehn regions (Streiff-Becker, 1947). As a consequence, the Foehn locations closest to the crest can be located within the region of precipitation and the Foehn region is shifted northward (Fig. S5f). Nevertheless, it needs to be stressed that the overall very low Foehn frequencies for the stations far away from the Alpine crest (see Table A1) make the results less robust, especially due to the limited time period of five years considered in this study.

In summary, the spatial variability of Foehn occurrence is, on the one hand, governed by the distance and elevation difference to the Alpine crest. On the other hand, local effects likely modulate the occurrence frequencies as well, and the different characteristics of the Foehn types likewise lead to regional variations in Foehn occurrence.

## 7 Discussion

The Foehn classification presented in this study builds upon the long record of research on the Alpine South Foehn. The body of literature comprises an extensive list of different Foehn types, including, for example, 'anticyclonic Foehn', 'cyclonic Foehn', 'dimmerfoehn', or 'shallow Foehn'. In this regard, several aspects are discussed in order to highlight similarities as well as discrepancies of our Foehn types to the existing literature:

– First of all, Foehn onset can occur while the Alpine region is still under the influence of an anticyclone and subsidence prevails above an elevated inversion layer. But does our *Dry Foehn* subtype correspond to the 'anticylonic Foehn'? An answer to this question is not straightforward, not least because the definition of the latter is rather vague in the literature. Billwiller, according to Kuhn (1984), considered the large-scale subsidence during this initial stage as a type of Foehn wind, a view rightfully challenged by Gubser (2006). At the same time, 'anticyclonic Foehn' is also described as a Foehn

where geopotential above the Alps is higher than usual, the isohypses are curved anticyclonically, and the Alpine south side completely lacks precipitation. Sometimes, even cloud-free conditions prevail (e.g., Burri et al., 1999; Gerstgrasser, 2017). This characteristic of 'anticyclonic Foehn' is also shared with our *Dry Foehn*. Another analogy can be drawn with respect to the 'Austrian Foehn' type, which occurs in absence of upstream precipitation, yet blocked cold air decouples the Po Valley boundary layer from the southerly flow aloft (Steinacker, 2006). Again, *Dry Foehn* characteristics as elaborated in Section 4 match with the 'Austrian Foehn' type.

– Analogously, the main 'cyclonic stage' of a typical Foehn event according to Billwiller, relates to our *Moist Foehn* subtype. During 'cyclonic Foehn' or the 'stationary stage' (Ficker, 1910), upstream orogaphic precipitation frequently forms and a Foehn wall establishes along the main Alpine crest (Frey, 1992). Likewise, the 'Swiss Foehn' type corresponds to our *Moist Foehn* and *Dimmer Foehn* subtypes, and to the 'trough Foehn' category of Burri et al. (1999).

– Diagnosing *Dimmer Foehn* from NWP data poses a particular challenge, since it is a local lee-side phenomenon. Hence, our classification should not be over-interpreted, as its definition is based on the overall spillover of precipitation beyond the crest in the Central Alps, rather than the actual presence of *Dimmer Foehn* within the Reuss Valley at Altdorf. Still, the mean synoptic conditions correspond to properties emerging from case studies in the literature (e.g., Streiff-Becker, 1947), as for example the low-pressure system being closer to the Alps, and stronger crest-level winds. These correspondences provide confidence in the diagnostic applied to classify *Dimmer Foehn*.

– It is notable that the different *Deep Foehn* subtypes (*Dry Foehn*, *Moist Foehn*, *Dimmer Foehn*) are oftentimes considered different stages during the typical temporal evolution of a Foehn event (e.g., Ficker, 1910; Streiff-Becker, 1933). Our subtypes resemble, to some extent, these stages ('anticyclonic Foehn', 'cyclonic Foehn', 'dimmerfoehn'), as described for example by Streiff-Becker (see Kuhn, 1984). However, as the study classifies Foehn hours instead of Foehn events, it remains to be investigated how often the subtypes transition from one to the other within the same event.

– The occurrence frequency of *Shallow Foehn* (6.3%) is comparable to the value reported in Seibert (1990) for the Austrian Brenner transect (10%). Furthermore, the analysis confirms the gap flow characteristics of *Shallow Foehn* as analysed during MAP (Mayr et al., 2007). The gap flow is driven by a colder air on the Alpine south side, inducing a northward hydrostatic pressure gradient force (Mayr and Armi, 2008). A rather puzzling finding concerns the weakly stratified boundary layer over the Po Valley during *Shallow Foehn*. In fact, this might be an effect of seasonality rather than a property related to the Foehn type itself. During the analysis period of five years, *Shallow Foehn* predominantly occured during spring (87.1%). At the same time, Würsch and Sprenger (2015) detected a relatively weak upstream stratification over the Po Valley for Foehn in Altdorf during spring. Whether the weak stratification links to the *Shallow Foehn* type or the seasonality could be addressed in a future study using a longer timeseries.

– As mentioned in the introduction, *Gegenstrom Foehn* could also be regarded as a special type of *Shallow Foehn* (e.g., Seibert, 1990). However, due to the strikingly different synoptic conditions, i.e. very strong and zonal large-scale flow, we consider it justified to define a separate category for such events. Furthermore, additional processes, besides north-south

temperature differences, might magnify the pressure gradient during this Foehn type. This includes the rapid pressure drop related to an approaching cyclone on the Alpine north side (Güller, 1977), and the pronounced synoptic pressure gradient. While the enhanced north-south pressure difference identified in Section 6 corroborates this hypothesis, more process-based studies would certainly be beneficial to better understand this exceptionally rare Foehn type.

Recent studies on the Alpine Foehn frequently apply Lagrangian diagnostics to study the evolution of air parcels prior to their arrival in the northern Foehn valleys (Würsch and Sprenger, 2015; Miltenberger et al., 2016; Saigger and Gohm, 2022; Jansing and Sprenger, 2022). Hence, a few aspects of our Lagrangian analysis are discussed and an outlook to potential future studies is given:

- On the one hand, air parcels during *Dry Foehn* originate from higher elevations upwind, which matches the first studied case of Miltenberger et al. (2016) and supports isentropic drawdown to be the main Foehn air warming mechanism for this *Deep Foehn* subtype. However, air parcels during *Moist Foehn* and *Dimmer Foehn* originate from significantly lower levels. This, on the other hand, is consistent with the thermodynamic theory according to Hann (1866), where Foehn air parcels are diabatically heated during their upstream ascent towards the Alpine crest. Overall, in agreement with Jansing and Sprenger (2022), the variability in the upstream ascent points towards a more balanced view on Foehn air warming, where different mechanisms can govern the Foehn-induced temperature increase, depending on the event. Besides, even for a certain subtype (e.g., *Moist Foehn*) trajectories can originate from various levels upwind, and not all of them ascend steeply towards the crest. Thererfore, a detailed heat budget analysis for a multitude of Foehn events and types needs to be conducted in order to increase the certainty with respect to the findings of Section 5. Besides, the trajectory analysis can be re-linked to a question posed in Seibert (1990) with respect to the fate of the low-level easterlies in the western Po Valley. With our analysis, we reveal that a part of this easterly flow actually ascends to contribute to the flow in Altdorf during *Moist Foehn* and *Dimmer Foehn*. A future study will revisit the lifting mechanism that provides these air parcels the ability to ascend from the boundary layer. Especially, it will be of interest whether the presence of a low-level temperature gradient in the Po Valley can facilitate the lifting via isentropic upglide.

- In comparison to Würsch and Sprenger (2015), besides *Dry Foehn*, also *Shallow Foehn* exhibits 'Austrian Foehn' characteristics. More precisely, this means air parcels stem from an elevated source region, loose little to no moisture during their approach to the Alpine crest, and are associated with negligible amounts of precipitation. Furthermore, all of the Foehn types not (or only weakly) associated with precipitation (*Dry Foehn*, *Shallow Foehn*, *Gegenstrom Foehn*) exhibit a pronounced stable layer at approximately crest level, which confirms the description of Seibert (1990) for a typical 'Austrian Foehn' event. At the same time, *Moist Foehn* and *Dimmer Foehn* correspond to the 'Swiss Foehn' type. This, however, illustrates that the 'Swiss Foehn' category of Würsch and Sprenger (2015) contains events with strikingly different characteristics. Hence, it needs to be stressed that not even both, but multiple Foehn types can occur at the same location depending on the driving synoptic conditions. To some extent, this has already been confirmed by Würsch and Sprenger (2015) with regard to the large spread in the upstream properties (e.g., altitude) of air parcels as identified by their Lagrangian analysis.

- Some of the findings with respect to the Lagrangian analysis cannot be entirely explained by the study. First of all, it seems that the very pronounced stable layer near crest level constitutes a less decisive factor for the vertical sourcing of air during *Gegenstrom Foehn*. It remains, however, unclear what the lifting mechanisms are in such cases. The authors can only speculate upon the potential role of the western flank of the Western Alps and the pronounced north-south baroclinicity across the Alps. Further investigations are needed to understand the mechanisms acting to lift air parcels during such cases. Additionally, the physical processes causing the humidity increase of cluster 3 trajectories are still to be determined explicitly. Interestingly, Miltenberger et al. (2016) found slightly increasing humidity values upstream of the crest in their *Dry Foehn* case as well, which, at least, provides additional confidence in our result. However, for a more in-depth study of the turbulent mixing process inducing the moisture gain, Lagrangian particle dispersion models would be better suited compared to kinematic air parcel trajectories used in this study. The same likewise holds true for a more exact quantification of the input of air parcels from the Po Valley mixed layer below the strongly stratified layer during *Shallow Foehn*.

## 8 Conclusions

This study aims at distinguishing different types of Alpine South Foehn mentioned in the literature, as well as characterising them from the synoptic to the Alpine scale. Furthermore, emphasis is given to their local meteorological impact at Altdorf, a location within a major Swiss Foehn valley on the Alpine north side, and other Foehn locations. To this end, we employ an observational-based Foehn index to define a Foehn timeseries for Altdorf. The timeseries, covering five years of Foehn occurrence (Nov 2015–Nov 2020), accumulates to a total of 2'329 Foehn hours that are analysed with the aid of operational COSMO analyses. The high resolution of the analysis data (1.1 km spatial and 1 h temporal) allows us to scrutinize the mean conditions of the different Foehn types on the Alpine scale as well as to investigate the origin of the air and the upstream thermodynamics from a Lagrangian point of view. The key results are summarized as follows:

- A subjective decision tree using features extracted from the COSMO-1 analysis (e.g., mean wind speed in a circular region around Altdorf at 700 hPa) serves to classify each Foehn hour as one of three main Foehn types (*Deep Foehn*, *Shallow Foehn*, *Gegenstrom Foehn*) and, if belonging to the *Deep Foehn* type, three respective subtypes (*Dry Foehn*, *Moist Foehn*, *Dimmer Foehn*). The main Foehn types are associated with strongly contrasting synoptic conditions. While *Deep Foehn* occurs downstream of an upper-level trough, *Shallow Foehn* is related to calm winds above crest level (weak synoptic forcing). *Gegenstrom Foehn* occurs during strong zonal large-scale flow within a region of pronounced synoptic pressure gradients over Europe. The *Deep Foehn* subtypes, in turn, differ with respect to the elongation and depth of the upstream upper-level trough. A more eastward trough axis and a deeper trough tends to induce stronger and more widespread upstream precipitation, with spillover beyond the Alpine crest.

- The differences in the synoptic conditions are also reflected in contrasting mesoscale conditions across the Alps. The deep layer of southwesterlies impinging on the Alpine barrier during *Deep Foehn* induces orographic lifting and precipitation

on the upwind side. Meanwhile, the descending air, potentially related to vertically propagating gravity waves, leads to a relative drying in the lee-side Foehn valley. No noteworthy orographic effect except flow channelling is present during *Shallow Foehn*. The colder air on the south side of the Alps induces Foehn as a hydraulic compensation flow between the two air masses on both sides of the Alps. As such, it is restricted to major gaps (e.g., Gotthard pass) along the Alpine transect. For *Gegenstrom Foehn*, the northward component of the flow is confined to the major Alpine north-south gaps as well. Above crest level, strong zonal winds advect moist air masses from the west, thereby reducing the drying effect of the descending Foehn air on the lee side. Among the *Deep Foehn* subtypes, *Dry Foehn* emerges as the one associated with dry conditions above crest level, potentially related to anticyclonic subsidence. Within the lee-side Foehn valley, relative humidity values likewise drop to the lowest of all Foehn types. While the conditions during *Moist Foehn* largely resemble the *Deep Foehn* type, a deep moistening with intense orographic precipitation is observed upwind of the Alps during *Dimmer Foehn*. The partial spillover of precipitation into the usually dry Foehn regions on the north side leads to elevated relative humidity values in the Foehn valley.

- Lagrangian backward trajectories, calculated for each of the Foehn hours, are classified into three clusters using a PCA-based metric. The three resulting clusters are linked to a different air mass origin and, accordingly, a distinct thermodynamic evolution. Cluster 1 trajectories originate over the central to eastern Po Valley at an altitude of 1.5 km. Hence, they ascend considerably during their anticyclonically curved approach to the Alpine crest (at 3 km altitude), experience pronounced latent heating in clouds ($\Delta\Theta$ ~4.5 K), and constitute the main precipitating airstream of Foehn air parcels arriving in Altdorf. Trajectories in cluster 2 originate at the Ligurian coast at a mean altitude of 2.5 km. Hence, they experience weaker latent heating ($\Delta\Theta$ ~1 K) as compared to cluster 1 trajectories during their slight ascent and, therefore, cluster 2 is associated with little precipitation. The characteristics of cluster 3 strongly contrast those of the other two clusters: Originating from above crest level (3.5 km), the air parcels are advected with the southwesterly large-scale flow towards the crest upstream of Altdorf. These air parcels descend towards Alpine crest level and are subject to a diabatic cooling of 4 K.

- As cluster 1 is the main precipitating airstream, it is especially important for *Moist Foehn* and *Dimmer Foehn*. Clusters 1 and 2, on the other hand, are more important for *Dry Foehn*, *Shallow Foehn* and *Gegenstrom Foehn*. Cluster 3 is most important in the latter case, as air parcels stem from the most western region during *Gegenstrom Foehn*. The upstream stability profile is found to partially govern the vertical sourcing of Foehn air parcels. A pronounced stable layer confines most of the Foehn air parcels to originate from regions above 2.5 km during *Dry Foehn*, as it largely inhibits ascent from the Po Valley boundary layer. During *Moist Foehn* and *Dimmer Foehn*, where the stable layer is much less pronounced, more air parcels are able to ascend from levels below 2 km, i.e. cluster 1 becomes more prominent. The upstream stable layer during *Shallow Foehn* again limits the origin of most cluster 1 trajectories to above 3 km altitude. However, in contrast to *Dry Foehn*, cluster 2 trajectories partially originate from lower levels. This might be induced by upward mixing within the weakly stratified boundary layer in the Po Valley. While a very pronounced stable layer is present during *Gegenstrom Foehn*, it appears to be less decisive for the vertical origin of the Foehn air parcels. Potential

reasons include the fact that a major part of the air parcels originates on the western flanks of the Western Alps, and the pronounced northward slope of isentropes during *Gegenstrom Foehn*, which supports isentropic upglide during the approach of the crest from the south.

- The presence of different Foehn types exhibits a profound impact on the local conditions at Altdorf as diagnosed by station measurements. While the mean wind and the gusts are of similar magnitude for the *Deep Foehn* subtypes (~40 $\mathrm{km\,h^{-1}}$ and ~60 $\mathrm{km\,h^{-1}}$, respectively), relative humidity varies between below 20% for *Dry Foehn* and above 80% for *Dimmer Foehn*. Regarding the Foehn-induced temperature increase, *Dry Foehn* and *Moist Foehn* exhibit peak values (median of 7.5–8 K), whereas the warming is slightly less pronounced for *Dimmer Foehn* (6.5 K). *Shallow Foehn* emerges as the weakest Foehn flow (mean wind of ~25 $\mathrm{km\,h^{-1}}$, gusts of ~40 $\mathrm{km\,h^{-1}}$), as the cross-Alpine pressure difference is also lower compared to the other types. The Foehn-induced warming is additionally reduced (4 K) for *Shallow Foehn* compared to the *Deep Foehn* subtypes. Mean winds (~32 $\mathrm{km\,h^{-1}}$) and gusts (~55 $\mathrm{km\,h^{-1}}$) of *Gegenstrom Foehn*, in turn, are only slightly weaker than during *Deep Foehn*. As during *Dimmer Foehn*, elevated relative humidity values are observed (median above 40%), while the warming is likewise less pronounced (4 K).

- The Foehn frequency at Swiss Foehn stations exhibits pronounced variability if a certain Foehn type prevails in Altdorf. First of all, Foehn frequencies generally decrease as one moves further down-valley (e.g., Evionnaz to Aigle). They also decrease with increasing horizontal distance to the Alpine crest (e.g., Altdorf to Einsiedeln), as well as with increasing altitude difference to major upstream gaps and the respective crest (e.g., Crans-Montana to Sion). Secondly, specific topographic peculiarities considerably determine the likelihood with which a certain Foehn type occurs at particular stations. As an example, the Foehn flow is not able to break through in Adelboden when *Shallow Foehn* prevails in Altdorf because the upstream mountain range is too high. Finally, when *Dimmer Foehn* is detected in Altdorf, the probability for the Foehn flow to reach the Swiss Plateau stations farther north (e.g., Zurich) is potentially increased, although one would need a longer timeseries to further corroborate this.

In this study we focused on Foehn occurring in Altdorf. To what extent our findings apply to other Foehn locations remains for future research. As the Foehn types are based upon different synoptic weather situations, they are likely to co-occur at multiple locations. In contrast, upstream properties of air parcels are expected to differ substantially. Both Würsch and Sprenger (2015) and Jansing and Sprenger (2022) showed that there exists pronounced valley-to-valley variability, with a much stronger influence of the Po Valley jet for Foehn winds in the Western Alps. Hence, the outcome of the Lagrangian analysis is likely to change if conducted, for example, for the more eastern Rhine Valley.

Furthermore, some caveats are discussed with respect to the data set used in this study. Despite the high resolution, the operational analysis can, similarly to forecasts (Wilhelm et al., 2012; Sandner, 2020), exhibit distinct model biases. For a detailed analysis of these biases, the reader is referred to Tian et al. (2022). Nevertheless, we do not expect near-surface model deficiencies to greatly impact our results, as we mainly focus on mesoscale aspects and refrain from an explicit consideration of local properties in Foehn valleys (e.g., at Altdorf). For the latter, we employ observations instead. With respect to the local-scale impact, a preliminary analysis has been conducted revealing considerable differences in the conditions at Altdorf depending on

the prevailing Foehn type, as well as an impact on the Foehn occurrence frequency at other stations in Switzerland (Section 6). The findings largely correspond to expectations from a forecaster's perspective. In order to explicitly quantify the occurrence of Foehn types at all the stations considered, one would either need additional analysis data, or use the new observational-based Foehn index instead (see Section 2.1). Furthermore, future studies could re-assess the linkage of Foehn types at multiple stations by allowing for a temporal lag between them rather than searching for exact overlaps in the occurrence of Foehn.

Additionally, a side note with respect to the Foehn types, as classified in this study, is made. Since the motivation for the different Foehn types arises from the comprehensive existing literature, it is based upon a subjective decision tree. However, the contrasting synoptic conditions of the different Foehn types give confidence in the validity of the methodological approach. The results could be compared to a statistical method, as for example self-organizing maps, an approach chosen by Kusaka et al. (2021) to classify South Foehn in Japan. Naturally, the defined Foehn types are not all-encompassing. Further differentiation between pre-frontal Foehn events and trough Foehn events according to Burri et al. (1999) could be considered. Besides, there exists a rich palette of local Foehn flavors not in the focus of the study. For example, Streiff-Becker (1947) differentiates between different Foehn types for the Canton of Glarus ('Kleintal- and Grosstalfoehn'). Additional, local Foehn winds include the 'Guggifoehn' in the Bernese Alps (Gerstgrasser, 2017), Westfoehn in Lucerne or Eastern Switzerland (Krieger et al., 2018), or the Pfaenderwind at the exit of the Rhine Valley (Gohm et al., 2015).

In summary, the first climatological assessment of Alpine South Foehn types reveals the existence of strikingly different Foehn types, differing with respect to their synoptic to mesoscale characteristics, as well as the upstream air parcel properties. The Foehn types also produce a distinct fingerprint in the local lee-side conditions, a finding valuable for forecasters. Still, open questions with respect to the processes inducing the warming or the mechanisms associated with the descent of the Foehn air remain to be investigated. In a future study, these questions will be tackled using a palette of high-resolution simulations representing all of the different Foehn types.

*Code and data availability.* Operational MeteoSwiss analyses, as well as measurement data, are available for research purposes upon request to MeteoSwiss. Processed data are available from the authors upon request. The trajectories have been compiled using LAGRANTO (Wernli and Davies, 1997; Sprenger and Wernli, 2015). The code used for the analysis and visualization is written in Python 3.9 and available from the authors upon request.

*Author contributions.* LJ, MS and LP equally contributed to the design of the study and the interpretation of the results. LJ extracted the analysis data from the MeteoSwiss archive, prepared it for the study, and conducted the analysis including the figures in the context of his PhD project. LJ wrote the paper, supported by MS and LP. BD drafted the methods section related to the Foehn index. DG provided input from a forecaster's perspective. All co-authors provided feedback on the paper.

*Competing interests.* The authors declare that no competing interests are present.

*Acknowledgements.* The Swiss National Science Foundation is acknowledged for the funding (grant nr 181992). We want to thank the Federal Office of Meteorology and Climatology MeteoSwiss for providing access to their data archive and the resources used for the data extraction. In particular, Marco Arpagaus and Daniel Leuenberger from MeteoSwiss are acknowledged for additional information related to the COSMO analysis data. In addition, we want to express our thanks to Heini Wernli and Marco Stoll for interesting discussions on several aspects of the study. Besides, we are grateful to our project partners Juerg Schmidli and Yue Tian from Goethe University Frankfurt (GUF) for providing helpful comments during our project meetings, which helped to improve the outcome of the study.

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
