# Peer review of "Classification of Alpine South Foehn based on five years of km-scale analysis data"

_Weather and Climate Dynamics, 2022_

## Referee Comment (RC1)

**Review of "Classification of Alpine South Foehn based on five years of km-scale analysis data" by Jansing et al.**

The authors present a climatological analysis of the large-scale conditions, thermodynamic history of foehn air parcels, and local conditions in Altdorf for five subjectively defined (often used, not well-defined in literature) foehn types. The analysis is based on station data and COSMO analysis data at a resolution of 1 km for 2016-2021. The results to a large degree justify the subjective categorisation by revealing distinctly different synoptic conditions and meso-/local-scale characteristics. The paper is generally very well written and if the following comments (largely pertaining to the presentation not the contents of the paper) are addressed, I recommend the paper for publication.

**Major comments**

1. l. 417: Why use 6 h travel time and not a particular distance from crest? Travel time will place the "upstream" location at very different distances from crest for parcels travelling in the PBL compared to those in the free troposphere (and for cases with varying free tropospheric wind speeds).

2. l. 620: Is it possible to explain differences in local meteorological changes in Altdorf, which are due to differences between the pre-foehn valley atmosphere and meteorological properties of the foehn air, solely with the thermodynamic pathway of foehn air parcels? Given the different synoptic conditions, would it also be a possibility that temperature and humidity (structure) of the pre-foehn valley atmosphere varies between the different types?

3. Section 6.2: Consider removing this section from the paper. The paper is already quite long with all the other interesting analysis, this section has methodological issues (everything being pre-conditioned on Foehn occurrence in Altdorf), and overall does not provide a lot of new insight.

4. Conclusions: The bullet points with the key findings are very lengthy and often are a mixture of methods and results. Would it be maybe more useful to have one bullet point per foehn type summarising all the key characteristics of that type (large-scale, meso-/local scale, Lagrangian) and a separate paragraph summing the Lagrangian results (cluster analysis, before tying them to the foehn types)?

**Minor comments**

1. l. 6: "Mean wind direction and speed around Altdorf" - please provide the spatial scale of averaging as the statement as is can be misleading (local effects?)

2. l. 57: briefly explain the term "foehn wall"

3. l. 180 f: The explanation of "m" is not understandable without consulting Duerr (2008). Please expand or rephrase.

4. l. 221ff: By starting the back trajectories in Altdorf you implicitly assume that the first part of the travel to the Alpine crest is captured correctly. Please comment on this.

5. Figures: In many colourbar labels missing (variable!), e.g. "geopotential height" in Fig. 3.

6. l. 382ff / Fig. 4: The relative humidity values in the "dry foehn" composite seem to be comparable to those in the "shallow foehn" composite, also there seems to be a stable layer over the Po valley during "dry foehn". Please comment.

7. l. 421f: Would be interesting to see how the scatter looks in the original phase-space. Maybe include scatterplots in the SI.

8. l. 421: Explain how variables are standardised.

9. l. 435: Linking the cluster Δlon to a specific geographic location is possible, because trajectories cross the crest at very similar locations? Please comment.

10. l. 459: "their ascent speed increases" > You infer this from the spacing of points along the T axis? If yes, then this is also the case for cluster 1 and should be mentioned in the discussion there as well.

11. l. 487: "mainly associated with cloud formation and only minor amounts of precipitation" - Why would that be? Can you speculate on the processes leading to inefficient precipitation formation in these cases?

12. Fig. 9: Why do the largest moisture losses occur so much before the main precipitation region? Also somehow indicate the areas blanked out due to small sample size.

13. l. 578f: "constrain the observed wind speed" - Is a range of 10-60 km/h not almost the full climatological range to be expected? I do not understand the word "constrain" in this context.

14. l. 600f: Is this a consequence of "unusually" high temperatures or "unusually" low specific humidity and what is considered as "usual" values (the non-foehn climatology)?
(This question also applies to the humidity signal of the other foehn types discussed in the following)

15. l. 815f: "distinct thermodynamic evolution" - The clear differences only arise when considering the mean, or? If considering all trajectories (Fig. 6) there seems to be rather a continuum of different properties.

**Technical corrections**

1. l. 32f: "relative drying of the air" - specify relative to which airmass.

2. l. 53: "while foehn winds **are** already"

3. l. 60f: "Another example" - what is the first example?

4. l. 63: "where **a** record-breaking cross-Alpine differen**ce** of 24hPa **was** observed"

5. l. 67: "humid air precipitates" - You mean precipitation forming in the still (on the Alpine North side) very humid and cloudy foehn airstream falls into the valley atmosphere? Please reformulate.

6. l. 75: "To this point" > "To this point in time" or "So far"

7. l. 115: "The other Foehn focus of MAP" > "Another focus of MAP"

8. l. 135: "these type**s** of upstream airstreams" (?) - Also not clear what types you are referring to.

9. l. 140: remove "Especially"

10. l. 188: Please cite the more up-to-date reference of Baldauf et al. (2011)
    @article{Baldauf2011,
    author = {Baldauf, M. and Seifert, A. and F\"orstner, J. and Majewski, D. and Raschendorfer, M.},
    journal = {Monthly Weather Review},
    pages = {3887-3905},
    title = {Operational convective-scale numerical weather prediction with the COSMO Model: Description and Sensitivities},
    volume = {139},
    year = {2011}}

11. l. 189: "Aside from NWP at the convective scale" - not clear what you mean here with NWP (conventional meaning does not make a lot of sense in this context)

12. l. 199f: Sounds like this has already been done at coarser spatial resolution

13. l. 261: "The procedure **is as** follows"

14. l. 283: "**are comparable** to Innsbruck"

15. l. 294: "approach in **our** study"

16. l. 321: "albeit **a** more northerly flow … in **his case** study" (is there another than this case study from Gueller?)

17. l. 344: "**see Fig.** S5a"

18. l. 356: "**as is** typical for"

19. p. 14, footnote: "the mean potential **temperature** at 3 km altitude"

20. l. 368: "field **at** 2.5 km"

21. l. 368f: "discussing the **correlation of** the Altdorf foehn type **with** the Foehn"

22. l. 370: "to a **weak** orographic lifting with **little** precipitation"

23. l. 371/384: "moistening" implies an increase in specific humidity, but what you show is an increase in relative humidity.

24. l. 396: "structure (**relative humidity** distribution, precipitation"

25. l. 405: "pathway. **In this section we** will"

26. l. 406: "their occurrence" - unclear reference of "their"

27. l. 407: The "linkage" of what?

28. l. 401: "and **to** quantify"

29. l. 421: "**For further** analysis"

30. l. 426: "correlated **with** the"

31. l. 434: "This **is evident** from" (?)

32. l. 439: "exhibit **a larger** spread"

33. l. 440: "corresponding **variability** in the"

34. l. 463: What do you mean with "a particularly northern position"?

35. l. 492ff: Are these parcel passing close to the cloud top? What about longwave cooling?

36. l. 526: "stable layer that largely inhibits the ascent of air parcels" - What role does the limit on trajectory travel time of 6h play for this conclusion?

37. l. 537: "(Fig. 11d). **The** majority"

38. Fig. 11: Mention in the caption that these are E-W cross-sections.

39. l. 552: "(Fig. 4c) **indicates** adiabatic ascent" (?)

40. l. 554: "while the trajectories **originating** from above"

41. l. 617: "spread **ranging from** below-zero"

42. l. 645: "valleys. **This** section"

43. l. 770: "explained by **our** study"

44. l. 783: "distinguishing" seems not to be the right word here - maybe "formalising" or "better characterising"?

45. l. 799: "precipitation, with spillover"

46. l. 800: "Alps. **A** deep"

---

## Referee Comment (RC2)

**Review of** "Classification of Alpine South Foehn based on five years of km-scale analysis data", by Lukas Jansing et al.

The authors present a climatological analysis and classification of Alpine South Foehn, focusing on Altdorf, Switzerland, which is one of the Alpine-wide "hot spots" for the occurrence of foehn. The study takes benefit from a five-year history of operational 1-km COSMO analysis data generated at MeteoSwiss, providing a reasonable resolution of local foehn characteristics at least in some larger Alpine valleys. I agree with Reviewer #1, who happened to be slightly faster than me, that the paper is very well written and should be published after some minor revisions. In the subsequent list of specific comments, issues already raised by Reviewer #1 will not be repeated.

**Specific comments**

The most surprising result for me is the weak stratification on the Alpine South side during Shallow Foehn, remembering that most of the Shallow Foehn Cases observed and investigated during MAP were characterized by very high stability over the Po Valley. Although this aspect is briefly discussed on page 32, it would be interesting to elaborate a bit more on this apparent discrepancy: How many the MAP Shallow Foehn cases would be classified as Shallow and Gegenstrom, respectively, according to the criteria applied in this study? If the majority is indeed Shallow, then the seasonality argument would be corroborated.

p. 17, discussion of Dimmerfoehn: I remember from the Dimmerfoehn event of 14–16 November 2002 that phases of particularly strong precipitation spillover were preceded by shallow cold front passages north of the Alps, reducing the lee-side subsidence related to the foehn flow and temporarily interrupting the foehn at low-elevation sites like Altdorf. It would be interesting to know if this type of interaction was just specific to this case (which was unusual in several aspects) or can be observed in a significant fraction of Dimmerfoehn events. Obviously, a far penetration of the foehn flow into the Alpine foreland will be inhibited by such cold front passages, and the variability of the foehn penetration distance may be correlated with or even controlled by the presence of low-level cold air in the foreland.

p. 20/22, discussion of diabatic cooling for cluster 3 trajectories: it appears quite surprising that diabatic cooling can keep subsiding trajectories at approximately constant temperature. Evaporation of precipitation entering from above would at best be able to maintain a moist adiabatic gradient, and more fundamentally, frontal lifting would typically not be associated with a dry layer between altitudes of 3 and 4 km. Likewise, the presence of strong turbulent vertical mixing would require weak stratification, which in turn would render strong cooling along subsiding trajectories unlikely. My hypothesis would be that the accuracy of the cluster 3 trajectories is not as good as for clusters 1 and 2. The latter pass over the Po Valley plain, whereas a significant fraction of the cluster 3 trajectories pass at low levels over the Alps, where the terrain-following coordinate surfaces are heavily distorted. The authors should include this aspect in their discussion. This also pertains to the middle paragraph on p. 35.

**Minor/editorial comments**

p. 6 / Eq. 1: $UU$ is quite an unusual notation for relative humidity. Why not using $RH$?

p. 11 / l. 290: "overlap of Dimmer Foehn and Gegenstrom Foehn hours ..." might be misunderstood in the way that the classification yields ambiguous results for these foehn types. "match" would fit better than "overlap"

Caption of fig. 5: I think it should read "Arrows pointing to the *right* correspond to eastward winds".

l. 464, last word: "effect" → "affect"?

l. 537: I don't see any cluster 1 signatures in Fig. 11d above 3 km. Should it read cluster 3?

l. 571–579: To explain the reduced wind speeds for Dimmerfoehn, it could be mentioned in addition that precipitation spillover increases (by latent cooling) the leeside stratification and thus reduces the foehn penetration close to the Alpine crest.

l. 760: "loose" → "lose"

l. 825/826: "Clusters 1 and 2" should probably read "Clusters 2 and 3"

l. 832: same as for l. 537

---

## Author Comment (AC1)

**Final response to reviewers**
* * *
**Classification of Alpine South Foehn based on five years of km-scale analysis data**
**Lukas Jansing | Lukas Papritz | Bruno Dürr | Daniel Gerstgrasser | Michael Sprenger**
**Submitted to WCD, WCD-2022-24**
**August 4, 2022**
* * *
*We acknowledge the reviewers for their careful reading of our manuscript. They provide valuable comments and suggestions to improve it. In this document, we aim to answer to each of them (original reviewer comments in black, our answers in green, bold italics). We also mention the changes (including line numbers) we plan to make in the manuscript.*

**Reviewer 1:**

**Major comments:**
1. l. 417: Why use 6 h travel time and not a particular distance from crest? Travel time will place the "upstream" location at very different distances from crest for parcels travelling in the PBL compared to those in the free troposphere (and for cases with varying free tropospheric wind speeds).
   *Using a particular distance to crest as measure to define our variable set for the PCA would be feasible, if trajectories mainly crossed the Alps along a north-south axis. However, in our case, we identified several difficulties in doing so. First of all, a considerable share of air parcels deviates from the linear N-S pathway (e.g., air parcels in cluster 1 travel in more of a E-W direction over the Po Valley). Individual trajectories might even have more complex pathways (e.g., re-circulating or spiraling). Hence, they will travel for longer times and distances without reaching larger "distance to crest". Furthermore, some trajectories travel too slow to reach a reasonably large distance to crest (e.g., 300 km) during their life span. Other air parcels might reach the distance to crest, however within a much larger time period, which leads to the fact that their Lagrangian history is much longer compared to other air parcels. These reasons render "distance to crest" a, in our opinion, rather unfeasible measure to extract the relevant set of variables.*

   *Furthermore, by using a certain time period as measure (instead of distance to crest), we give each air parcel the same time period during which it is subject to physical processes (e.g., latent heating in clouds) on its pathway to the Alpine crest. By this, we allow for a fair comparison between them, however accepting that the distances to crest might be very different. Another option would be to consider the integrated travel distance of trajectories to crest. However, this approach would likely deliver similar results to just considering the 6 h time difference. To further corroborate this, we extracted a corresponding variable set based on 300 km upstream travel distance to crest, performed a PCA and visualized the mean pathways of the resulting clusters (Fig. R1).*

   *As can be seen, the mean pathways emerge as overall very similar to the original classification. In this light, we would prefer to stick to our original version implemented. We*

*will add 1-2 sentences to L. 416 to better explain our reasoning for choosing travel time as the measure in order to compute our variable set for the PCA.*

[Figure]

*Figure R1. Same as Fig. 7a in the manuscript, but based on a PCA classification with a variable set extracted at 300 km integrated travel distance upstream of crest.*

2. l. 620: Is it possible to explain differences in local meteorological changes in Altdorf, which are due to differences between the pre-foehn valley atmosphere and meteorological properties of the foehn air, solely with the thermodynamic pathway of foehn air parcels? Given the different synoptic conditions, would it also be a possibility that temperature and humidity (structure) of the pre-foehn valley atmosphere varies between the different types?

*The reviewer raises a valid concern. The temperature difference between Foehn air and the air 3 h prior to Foehn breakthrough can depend upon properties of the Foehn air (and hence its Lagrangian history), as well as properties of the pre-Foehn valley atmosphere. For example, a more frequent occurrence of cold-air pools for a certain Foehn type might induce a larger temperature difference with respect to the Foehn flow, even if the thermodynamic history of the Foehn air parcels is similar for both of these Foehn types. Unfortunately, we cannot assess the vertical structure of the pre-Foehn valley atmosphere with measurements, as we do not have any sounding data from Altdorf. However, we evaluated the temperatures 3 h prior to Foehn breakthrough (used in Fig. 12d in the manuscript) to reveal differences between the Foehn types (Fig. R2).*

*As can be seen, there exist substantial differences between the different Foehn types. Temperatures prior to Foehn breakthrough tend to be lower for moist Foehn and Dimmer Foehn compared to Dry Foehn. Furthermore, temperatures are highest for Shallow Foehn and lowest for Gegenstrom Foehn. The higher pre-Foehn temperatures during Shallow Foehn could indicate that the pre-Foehn valley atmosphere during these cases is systematically warmer, which could indeed reduce the Foehn air warming (compared to the Deep Foehn categories), as displayed in Fig. 12 in the manuscript. However, the differences in pre-Foehn temperatures (and also the differences in Foehn temperatures) are also influenced by the different seasonality of the Foehn types (e.g., Shallow Foehn occurs more often during spring). Hence, we cannot perform a conclusive evaluation. However, we will include a few sentences in the manuscript discussing*

*this effect on the magnitude of the temperature jump during Foehn onset (at L. 632/633, after having discussed the temperature differences).*

[Figure]

*Figure R2. Boxplot of Altdorf temperature measurements. Shown are the temperatures 3 h prior to Foehn breakthrough for the different Foehn types. Note that the types are classified upon hourly basis, while the breakthrough time is identified by using an event definition (see manuscript L. 612-613).*

3. Section 6.2: Consider removing this section from the paper. The paper is already quite long with all the other interesting analysis, this section has methodological issues (everything being pre-conditioned on Foehn occurrence in Altdorf), and overall does not provide a lot of new insight.

*We thank the reviewer for this input and we also agree that the paper is rather long. Consequently, we will remove Section 6.2 from the manuscript and rename Section 6 to "Impact of Foehn types in Altdorf". However, we will include Fig. 13 in the supplement (as Fig. S8) and make a brief cross-reference to it in the Conclusions (L. 860). The removal of the Section will also lead to minor adjustments in Abstract, Introduction, and Conclusions.*

4. Conclusions: The bullet points with the key findings are very lengthy and often are a mixture of methods and results. Would it be maybe more useful to have one bullet point per foehn type summarising all the key characteristics of that type (large-scale, meso-/local scale, Lagrangian) and a separate paragraph summing the Lagrangian results (cluster analysis, before tying them to the foehn types)?

*Upon studying both the Discussion and Conclusions, we agree with the reviewer that the bullet points induce a feeling of lengthiness. We decided to restructure these Sections as follows:*

- *We will remove the bullet points in the Discussion and introduce two subsections for further structuring instead*
- *We will shorten the Conclusions a bit by removing some methodological details and some results (some shortening also occurs, because Section 6.2 will be discarded in the new version of the manuscript).*

*We will refrain from a restructuring according to the suggestion by the reviewer, since one bullet point per Foehn type would result in an even longer list compared to the current list (5 bullet points including all results).*

**Minor comments:**

1. l. 6: "Mean wind direction and speed around Altdorf" - please provide the spatial scale of averaging as the statement as is can be misleading (local effects?)

*We forgot to mention the spatial scale of the circle around Altdorf (radius of 100 km). We will mention it in the Abstract (L. 6) as well as in Section 3 (L. 241).*

2. l. 57: briefly explain the term "foehn wall"

*We will add a brief remark to L. 57.*

3. l. 180 f: The explanation of "m" is not understandable without consulting Duerr (2008). Please expand or rephrase.

*We will rephrase the respective sentence (at L. 180-181).*

4. l. 221ff: By starting the back trajectories in Altdorf you implicitly assume that the first part of the travel to the Alpine crest is captured correctly. Please comment on this.

*We partially agree with the reviewer's comment. We indeed implicitly assume the statistical distribution of positions (lon,lat,z) at crest level to be correctly captured by the backward trajectories. However, the pathway towards crest does not necessarily need to be correct for this assumption, as long as the distribution (lon,lat,z) at crest is reasonably represented. Besides, the trajectories cannot be completely erroneous, as they cross over the Alps from the south during most of the Foehn hours, which is the expected pattern for the examined South Foehn cases.*

*To substantiate our line of argument, we performed a little sensitivity case study for three Foehn days in Nov 2016, for which we have COSMO-1 hindcasts with 3D fields at higher temporal resolution (10-min; see also Jansing and Sprenger, 2022). We calculated hourly backward trajectories from 21 Nov 00 UTC – 23 Nov 23 UTC with 10 min input frequency as well as with 60 min input frequency of the wind fields. Then, we evaluated the difference in the position of the trajectories crossing the Alpine crest (Fig. R3).*

[Figure]

*Figure R3. 2D-histogram of intersection positions with the crestline of backward trajectories started around Altdorf (same starting setup as in manuscript). The trajectories are calculated for each hour from 21 Nov 00 UTC – 23 Nov 23 UTC using (a) hourly input fields, and (b) 10-min input fields. Intersections with the crestline below topography result from interpolation errors, as the positions are evaluated by linear interpolation between the two closest trajectory positions to the crestline.*

*As visible, the sixfold increase in temporal resolution of the input fields only leads to marginal shifts in the overall pattern. Thus, we assume our assumption to be valid for the COSMO-1 analysis backward trajectories. However, we will mention this assumption in Section 2.3 (L. 225).*

5.  Figures: In many colourbar labels missing (variable!), e.g. "geopotential height" in Fig. 3.
*Thanks for pointing this out. We will add the variable names to the colorbars.*

6.  l. 382ff / Fig. 4: The relative humidity values in the "dry foehn" composite seem to be comparable to those in the "shallow foehn" composite, also there seems to be a stable layer over the Po valley during "dry foehn". Please comment.
*Thank you for pointing this out. We agree that Dry Foehn and Shallow Foehn are similar in terms of the relative humidity values and the presence of the stable layer. We will add some sentences to the manuscript to highlight this correspondence of Dry Foehn to Shallow Foehn (at L. 385).*

7.  l. 421f: Would be interesting to see how the scatter looks in the original phase-space. Maybe include scatterplots in the SI.
*We include scatterplots of the variables in the original phase space in the final response document. However, we prefer not to include it in the supplement, as we do not discuss it.*

[Figure]

*Figure R4. Scatterplots of the variable set used for the PCA prior to standardization (see Section 5.1 for more details regarding the variable set).*

8.  l. 421: Explain how variables are standardised.
*They are standardized by subtracting the mean and dividing by the standard deviation for each of the six variables. We will mention it in Section 5.1 (at L. 421).*

9.  l. 435: Linking the cluster Δlon to a specific geographic location is possible, because trajectories cross the crest at very similar locations? Please comment.
*The Δlon is just used as part of the variable set to perform the PCA and subdivide our trajectories into three clusters. Afterwards, we again look at the actual trajectory positions (lon,lat,z) of each trajectory. Therefore, we can analyze the mean pathway of the different clusters.*

10. l. 459: "their ascent speed increases" > You infer this from the spacing of points along the T axis? If yes, then this is also the case for cluster 1 and should be mentioned in the discussion there as well.

*Yes, we infer it from the spacing of points along the T-axis. We will rephrase accordingly (at L. 460).*

11. l. 487: "mainly associated with cloud formation and only minor amounts of precipitation" - Why would that be? Can you speculate on the processes leading to inefficient precipitation formation in these cases?

*As this sentence is rather speculative (we did not specifically investigate cloud formation in the manuscript), we will rephrase it to address precipitation only (at L. 487-488).*

12. Fig. 9: Why do the largest moisture losses occur so much before the main precipitation region? Also somehow indicate the areas blanked out due to small sample size.

*We argue that the largest moisture losses do not happen so much before the main precipitation region. As Fig. 9 indicates specific humidity changes relative to crest, moisture loss happens where the contour field changes its values, so not south of Lago di Garda, but rather south of Lago Maggiore, which is where one of the main precipitation regions are situated as well.*

*We will indicate the areas blanked out owing to small sample size by brownish, transparent colors in Fig. S6 in the supplement and refer to it in the manuscript (at L. 475).*

13. l. 578f: "constrain the observed wind speed" - Is a range of 10-60 km/h not almost the full climatological range to be expected? I do not understand the word "constrain" in this context.

*We were referring to the similarities of the median wind speeds and the ranges of the boxes for the Deep Foehn subtypes. We rephrased accordingly (L. 578-579).*

14. l. 600f: Is this a consequence of "unusually" high temperatures or "unusually" low specific humidity and what is considered as "usual" values (the non-foehn climatology)?

*The usual values in this context are considered the "standard" Foehn conditions. In our case this would be either Deep Foehn, or, as we do not explicitly show Deep Foehn conditions in Fig. 12, the dominant Moist Foehn subtype. So, all of the statements addressing relative humidity in the section refer to this standard. We will slightly rephrase to make this clearer.*

*Whether the unusual low relative humidity values are a result of unusually high temperatures or unusually low specific humidity values is indeed a valid question, which we cannot answer with the analysis presented in this study. However, we can speculate upon it by comparing composite cross sections of both temperature and specific humidity for Dry Foehn and Moist Foehn (as an example; see Fig. R5).*

*Focusing on the upstream free-tropospheric atmosphere above the Po Valley (e.g., at 3.5 km altitude), we can see that during Dry Foehn, both temperature is higher and specific humidity is lower than during Moist Foehn, hence both contribute to the unusually low relative humidity. On the other hand, in the Foehn valley (see red dot), lower relative humidity values seem to be mainly associated with warmer temperatures.*

[Figure]

*Figure R5. As Fig. 4 in the manuscript, but for temperature and specific humidity during Dry Foehn (left column) and Moist Foehn (right column).*

15. (This question also applies to the humidity signal of the other foehn types discussed in the following)
*See above answer.*

16. l. 815f: "distinct thermodynamic evolution" - The clear differences only arise when considering the mean, or? If considering all trajectories (Fig. 6) there seems to be rather a continuum of different properties.
*Thanks for pointing this out. We will adjust the bullet point accordingly (L. 815-816) to highlight that we refer to the mean evolution.*

**Technical corrections:**
1. l. 32f: "relative drying of the air" - specify relative to which airmass.
*"Relative" referred to relative humidity. We will adjust the sentence.*

2. l. 53: "while foehn winds are already"
*We will correct it.*

3. l. 60f: "Another example" - what is the first example?
*There is no other example. We will rephrase.*

4. l. 63: "where a record-breaking cross-Alpine pressure difference of 24hPa was observed"
*We will correct it.*

5. l. 67: "humid air precipitates" - You mean precipitation forming in the still (on the Alpine North side) very humid and cloudy foehn airstream falls into the valley atmosphere? Please reformulate.
*We will reformulate the sentence.*

6. l. 75: "To this point" > "To this point in time" or "So far"

*We will adjust it.*

7. l. 115: "The other Foehn focus of MAP" > "Another focus of MAP"
*We will rephrase.*

8. l. 135: "these types of upstream airstreams" (?) - Also not clear what types you are referring to.
*We will correct it and slightly adjust the sentence.*

9. l. 140: remove "Especially"
*We will remove it.*

10. l. 188: Please cite the more up-to-date reference of Baldauf et al. (2011)
*We will add a reference to Baldauf et al. (2011) to L. 188.*

11. l. 189: "Aside from NWP at the convective scale" - not clear what you mean here with NWP (conventional meaning does not make a lot of sense in this context)
*We will rephrase the sentence accordingly.*

12. l. 199f: Sounds like this has already been done at coarser spatial resolution
*Thanks for pointing this out. We will rephrase.*

13. l. 261: "The procedure is as follows"
*We will correct it.*

14. l. 283: "are comparable to Innsbruck"
*We will correct it.*

15. l. 294: "approach in our study"
*We will correct it.*

16. l. 321: "albeit a more northerly flow ... in his case study" (is there another than this case study from Gueller?)
*We will adjust it. No, there is no other (published) study than the case study from Gueller.*

17. l. 344: "see Fig. S5a"
*We will correct it.*

18. l. 356: "as is typical for"
*We will correct it.*

19. p. 14, footnote: "the mean potential temperature at 3 km altitude"
*We will correct it.*

20. l. 368: "field at 2.5 km"
*We will correct it.*

21. l. 368f: "discussing the correlation of the Altdorf foehn type with the Foehn"

*The sentence is removed anyways since Section 6.2 will no longer be part of the manuscript.*

22. l. 370: "to a weak orographic lifting with little precipitation"
*We will adjust it.*

23. l. 371/384: "moistening" implies an increase in specific humidity, but what you show is an increase in relative humidity.
*Thanks for pointing this out. We will adjust it accordingly.*

24. l. 396: "structure (relative humidity distribution, precipitation"
*We will correct it.*

25. l. 405: "pathway. In this section we will"
*We will correct it.*

26. l. 406: "their occurrence" - unclear reference of "their"
*We will slightly rephrase.*

27. l. 407: The "linkage" of what?
*We were referring to the linkage of upwind airstreams to the Foehn types. We will rephrase.*

28. l. 401: "and to quantify"
*We will correct it.*

29. l. 421: "For further analysis"
*We will correct it.*

30. l. 426: "correlated with the"
*We will correct it.*

31. l. 434: "This is evident from" (?)
*We will correct it.*

32. l. 439: "exhibit a larger spread"
*We will correct it.*

33. l. 440: "corresponding variability in the"
*We will correct it*

34. l. 463: What do you mean with "a particularly northern position"?
**By this, we mean that the smooth crestline extends further north than the actual highest peaks in this region. We defined the crestline by searching for maximum topography along the model coordinates (rlon,rlat) within the Alpine region and subsequently smoothed the respective lines. We will rephrase the sentence to make it a bit clearer, however would like to refrain from describing the definition of the crestline in the manuscript, as it is rather technical.**

35. l. 492ff: Are these parcel passing close to the cloud top? What about longwave cooling?

*Thanks for mentioning this. Indeed, longwave cooling could be an additional mechanism inducing the diabatic cooling. We will include it in the text as possible process. However, we cannot assess whether the air parcels are close to the cloud top.*

36. l. 526: "stable layer that largely inhibits the ascent of air parcels" - What role does the limit on trajectory travel time of 6h play for this conclusion?

*This does not play any role for this conclusion. The limit of 6 h is only used to define the clusters using the PCA method. Afterwards, the complete trajectory data set is again used for further analysis (see also minor comment 9).*

37. l. 537: "(Fig. 11d). The majority"
*We will correct it.*

38. Fig. 11: Mention in the caption that these are E-W cross-sections.
*We will mention it in the caption.*

39. l. 552: "(Fig. 4c) indicates adiabatic ascent" (?)
*We will correct it.*

40. l. 554: "while the trajectories originating from above"
*We will correct it.*

41. l. 617: "spread ranging from below-zero"
*We will correct it.*

42. l. 645: "valleys. This section"
*This belongs to Section 6.2, which will be removed from the manuscript.*

43. l. 770: "explained by our study"
*We will correct it.*

44. l. 783: "distinguishing" seems not to be the right word here - maybe "formalising" or "better characterising"?
*We will use "formalising".*

45. l. 799: "precipitation, with spillover"
*We will correct it.*

46. l. 800: "Alps. A deep"
*We will correct it.*

**Reviewer 2:**

**Specific comments:**

1. The most surprising result for me is the weak stratification on the Alpine South side during Shallow Foehn, remembering that most of the Shallow Foehn Cases observed and investigated during MAP were characterized by very high stability over the Po Valley. Although this aspect is briefly discussed on page 32, it would be interesting to elaborate a bit more on this apparent discrepancy: How many the MAP Shallow Foehn cases would be classified as Shallow and Gegenstrom, respectively, according to the criteria applied in this study? If the majority is indeed Shallow, then the seasonality argument would be corroborated.

*We want to thank the reviewer for this interesting comment on the aspect of reduced vs. enhanced stability over the Po Valley during Shallow Foehn. Indeed, this was a surprising result for us as well, which we were not able to explain entirely. Furthermore, we are aware that, oftentimes, Shallow Foehn has been considered to occur under calm conditions above crest level, or when westerlies prevail (e.g., Gohm and Mayr, 2004). This might affect the outcome of our study related to the Po Valley stability, as we classify cases with westerlies above crest as Gegenstrom Foehn.*

*To shed some light onto the reviewer's question regarding the classification of MAP events, we decided to perform a simplified classification of the 12 MAP events using ERA-5 reanalysis data. We performed the classification as done in the manuscript, however restricted ourselves to the main Foehn types and used the wind field at 700 hPa and 500 hPa interpolated to Altdorf (no averaging within a circle). The outcome is depicted in Figure R6.*

[Figure]

*Figure R6. Classification of the MAP period (12 events; start- and enddates are taken from Drobinski et al., 2007). Each hour within an event is colored according to its associated Foehn type.*

*Evidently, most of the Foehn hours during the MAP period are classified as Deep Foehn, in alignment with the outcome of our study. Since there exists no MAP paper classifying all of the events into either deep or shallow, it is challenging to compare this result to the literature. Considering various MAP publications, the cases could be classified as follows:*

- *Event 1 (15 Sep): unclear, rather a Deep Foehn (Richner et al., 2006)*
- *Event 2 (19-20 Sep): Deep Foehn (Richner et al., 2006)*
- *Event 3 (22-23 Sep): Shallow Foehn (Richner et al., 2006)*
- *Event 4 (30 Sep): Shallow Foehn (Drobinski et al., 2003)*
- *Event 5 (02-03 Sep): Shallow Foehn (Drobinski et al., 2001)*
- *Event 6 (18 Oct): unclear, rather a Shallow Foehn (Richner et al., 2006)*
- *Event 7 (20-21 Oct): Shallow to Deep Foehn transition (Lothon et al., 2003)*
- *Event 8 (22-23 Oct): unclear, rather a Deep Foehn (Richner et al., 2006)*
- *Event 9 (24 Oct): Deep Foehn (Drobinki et al., 2006)*
- *Event 10 (30-31 Oct): Shallow to Deep Foehn transition (Drobinksi et al., 2003)*
- *Event 11 (01-02 Nov): Shallow Foehn (Richner et al., 2006)*
- *Event 12 (05-06 Nov): Deep Foehn (Drobinski et al., 2006)*

*The classification agrees well with MAP literature for the Deep Foehn events (events 2, 9, 12). There are some evident discrepancies for MAP Shallow Foehn events, where our classification in fact mostly indicates Deep Foehn (events 3, 4, 5). This can be explained by the fact that MAP studies classified some events with southwesterlies above crest level as Shallow Foehn, where our classification yields Deep Foehn owing to a wind direction below 240°. However, some Shallow Foehn events are classified correctly (event 6). Finally, as the reviewer mentioned, some time periods where MAP literature considers the Foehn flow to be shallow, turns out to be Gegenstrom Foehn in our classification (beginning of event 7, event 11).*

*In summary, some of the MAP cases (or periods of them) are indeed classified as shallow, however others are considered Deep Foehn events or Gegenstrom Foehn according to our criteria. Hence, we cannot fully corroborate the seasonality argument, but also not discard it. However, we will include a few sentences discussing the discrepancies to MAP results in the discussion (at L. 728f).*

2. p. 17, discussion of Dimmerfoehn: I remember from the Dimmerfoehn event of 14–16 November 2002 that phases of particularly strong precipitation spillover were preceded by shallow cold front passages north of the Alps, reducing the lee-side subsidence related to the foehn flow and temporarily interrupting the foehn at low-elevation sites like Altdorf. It would be interesting to know if this type of interaction was just specific to this case (which was unusual in several aspects) or can be observed in a significant fraction of Dimmerfoehn events. Obviously, a far penetration of the foehn flow into the Alpine foreland will be inhibited by such cold front passages, and the variability of the foehn penetration distance may be correlated with or even controlled by the presence of low-level cold air in the foreland.

*We agree with the reviewer that this is certainly an interesting interaction, which might be present during multiple Dimmer Foehn events. However, we have to refrain from a systematic analysis of it, since we are not able to objectively identify (shallow) cold fronts for all of the Foehn hours from the COSMO analysis. Nevertheless, we can provide the reviewer with a simplified analysis of low-level (700 m AMSL) temperature and wind speed composites of Dimmer Foehn comparing it to Moist Foehn hours (Fig. R7).*

*Dimmer Foehn is associated with lower temperatures on the Swiss Plateau, as well as in the typical Foehn valleys. However, the temperature gradient between the valleys and the foreland seems to be similar for both types. In turn, the mesoscale east-west temperature gradient is enhanced for Dimmer Foehn, which might indicate the more frequent presence of cold fronts during these events. Furthermore, it might also point towards the more frequent cold-air advection around the Western Alps during such cases (similar as in Zängl and Hornsteiner, 2007). However, focusing on the wind speed, we cannot observe a reduced penetration into the Alpine foreland during Dimmer Foehn. On the contrary, the penetration reaches a bit further into the foreland, whereas the southwesterlies north of the Alps are also enhanced.*

*These aspects of the passage of shallow cold fronts and the relation to Foehn penetration into the Alpine foreland during Dimmer Foehn would be an interesting aspect to discuss in Section 6.2. However, since we discard this Section, we will not discuss it further in the manuscript and leave it for future studies.*

[Figure]

*Figure R7. Horizontal composites at 700 m AMSL of temperature and horizontal wind for Moist Foehn (left column) and Dimmer Foehn (right column).*

3. p. 20/22, discussion of diabatic cooling for cluster 3 trajectories: it appears quite surprising that diabatic cooling can keep subsiding trajectories at approximately constant temperature. Evaporation of precipitation entering from above would at best be able to maintain a moist adiabatic gradient, and more fundamentally, frontal lifting would typically not be associated with a dry layer between altitudes of 3 and 4 km. Likewise, the presence of strong turbulent vertical mixing would require weak stratification, which in turn would render strong cooling along subsiding trajectories unlikely. My hypothesis would be that the accuracy of the cluster 3 trajectories is not as good as for clusters 1 and 2. The latter pass over the Po Valley plain, whereas a significant fraction of the cluster 3 trajectories pass at low levels over the Alps, where the terrain-following coordinate surfaces are heavily distorted. The authors should include this aspect in their discussion. This also pertains to the middle paragraph on p. 35.

*We thank the reviewer for pointing towards the incomplete discussion of the diabatic processes in Section 5.2. On the one hand, we argue that the moderate diabatic cooling (1.5 K) between 6 h and 2 h prior to crest could compensate the adiabatic warming, since the descent magnitude during this time period is also very small (see boxes in Fig. 7b and the small distance between dots in Fig. 8). As soon as the descent accelerates (during the last 2 hours), the temperature does in fact increase (despite further diabatic cooling). On the other hand, we agree with the reviewer that, following the logic in the comment above, it is unlikely that turbulent mixing acts to cool the subsiding air parcels. Besides, the accuracy of*

*trajectories in cluster 3 might indeed be reduced compared to the other two clusters. We conclude this not because of the heavily distorted coordinate surfaces over the complex terrain (LAGRANTO considers the terrain-following coordinates in its interpolation routine), but rather due to the fact that trajectories are generally less accurate over complex terrain because of the distorted and highly non-stationary flow (see also L. 221f in Section 2.3). We will discuss these aspects in Section 5.2 (turbulent mixing at L. 492f) and in the Discussion (diabatic processes and accuracy of cluster 3 trajectories at L. 770f).*

*Finally, we also want to point out that not all trajectories experience this diabatic cooling while subsiding at approximately constant temperature. This becomes evident by considering the vast spread in altitude of cluster 3 air parcels (Fig. 7b). In fact, Fig. 8 rather depicts the mean evolution, which can deviate substantially from the temporal evolution of single air parcels. We will try to highlight this more clearly throughout Section 5.2.*

**Minor/editorial comments:**

1. p. 6 / Eq. 1: UU is quite an unusual notation for relative humidity. Why not using RH?
*We will change it to RH.*

2. p. 11 / l. 290: "overlap of Dimmer Foehn and Gegenstrom Foehn hours ..." might be misunderstood in the way that the classification yields ambiguous results for these foehn types. "match" would fit better than "overlap"
*Thanks for this note, we will correct it.*

3. Caption of fig. 5: I think it should read "Arrows pointing to the right correspond to eastward winds".
*Indeed. Thanks for your careful reading, we will correct it.*

4. l. 464, last word: "effect" → "affect"?
*Yes, we will correct it.*

5. l. 537: I don't see any cluster 1 signatures in Fig. 11d above 3 km. Should it read cluster 3?
*Yes, it should read cluster 3. We will correct it.*

6. l. 571–579: To explain the reduced wind speeds for Dimmerfoehn, it could be mentioned in addition that precipitation spillover increases (by latent cooling) the leeside stratification and thus reduces the foehn penetration close to the Alpine crest.
*We will add two sentences mentioning this effect during Dimmer Foehn (at L. 579f).*

7. l. 760: "loose" → "lose"
*We will correct it.*

8. l. 825/826: "Clusters 1 and 2" should probably read "Clusters 2 and 3"
*We will correct it.*

9. l. 832: same as for l. 537
*Yes, again the same mistake. We will correct it.*